# Globally solving the Gromov-Wasserstein problem for point clouds in low dimensional Euclidean spaces

**Martin Ryner**
Vironova AB, Stockholm, Sweden
Division of Numerical Analysis, Optimization and Systems Theory,
Department of Mathematics,
KTH Royal Institute of Technology, Stockholm, Sweden
`martin.ryner@vironova.com, martinrr@kth.se`

**Jan Kronqvist**
Division of Numerical Analysis, Optimization and Systems Theory,
Department of Mathematics,
KTH Royal Institute of Technology, Stockholm, Sweden
`jankr@kth.se`

**Johan Karlsson**
Division of Numerical Analysis, Optimization and Systems Theory,
Department of Mathematics,
KTH Royal Institute of Technology, Stockholm, Sweden
`johan.karlsson@math.kth.se`

## Abstract

This paper presents a framework for computing the Gromov-Wasserstein problem between two sets of points in low dimensional spaces, where the discrepancy is the squared Euclidean norm. The Gromov-Wasserstein problem is a generalization of the optimal transport problem that finds the assignment between two sets preserving pairwise distances as much as possible. This can be used to quantify the similarity between two formations or shapes, a common problem in AI and machine learning. The problem can be formulated as a Quadratic Assignment Problem (QAP), which is in general computationally intractable even for small problems. Our framework addresses this challenge by reformulating the QAP as an optimization problem with a low-dimensional domain, leveraging the fact that the problem can be expressed as a concave quadratic optimization problem with low rank. The method scales well with the number of points, and it can be used to find the global solution for large-scale problems with thousands of points. We compare the computational complexity of our approach with state-of-the-art methods on synthetic problems and apply it to a near-symmetrical problem which is of particular interest in computational biology.

## 1   Introduction

Many important applications in machine learning deal with comparing sequences, images, and higher dimensional data, where the data is unstructured and not directly comparable. In physics, chemistry, biology, music, and linguistics, objects with greatly different properties often appear in symmetrical variations characterized by concepts such as isomerisms, chirality, harmonies, and alternations. Understanding, and being able to analyze, these types of variations can be truly critical

37th Conference on Neural Information Processing Systems (NeurIPS 2023).

as some variations in chemicals and biologicals may be toxic or even lethal. The Gromov-Wasserstein framework [15, 16, 17], has shown to be a powerful approach for comparing and matching such data, as it is invariant to translations and rotation. The Gromov-Wasserstein framework has, for example, been successfully applied to domain adaptation [29], graph matching [28], metric alignment [9], single-cell alignment [7], and word embedding [1].

The task of evaluating the Gromov-Wasserstein problem is in general considered to be intractable. Typically, the computational burden grows exponentially with the number of points describing the compared objects. In fact, a Gromov-Wasserstein problem can be formulated as quadratic assignment problem (QAP) [14, 5, 4], which is known to be NP-Hard. Naturally, there has been plenty of research on local and approximate methods for solving Gromov-Wasserstein and QAP problems [20, 23, 22, 27, 2, 24, 25]. However, objects containing symmetries or repeated patterns are particularly challenging for local optimization methods and may lead to significant errors in the estimated discrepancy as matching such objects with local optimization methods may accidentally find the sub-optimal reflections and rotations. The inability to detect such phenomena can have a great impact on the discovery of isomerisms and subsequently attributes of crucial importance.

In this paper, we develop a rigorous method for globally optimizing Gromov-Wasserstein problems by calculating a sequence of iteratively improving upper- and lower bounds. We consider a general class of Gromov-Wasserstein discrepancy problems where the points, representing the objects, belong to a Euclidean space. We show that such Gromov-Wasserstein problems can be formulated exactly as low-rank QAPs. We build upon this low-rank QAP representation to develop an algorithm that scales well with the number of points. The proposed algorithm can be characterized as a so-called cutting plane method [12, 10] where we solve a sequence of relaxed problems that are iteratively strengthened by generating and accumulating valid linear inequality constraints, *i.e.,* cutting planes. The optimum of the relaxed problem provides a valid lower bound for the optimum of the Gromov-Wasserstein problem in each iteration. By solving a computationally cheap optimal transportation problem [18, 26, 6], we obtain both an upper-bound and a new cutting plane to strengthen the relaxation. We prove convergence for the proposed algorithm, and present a computational study that clearly shows the algorithm's efficiency and that the performance scales well with the number of points.

The main contribution of the paper can be summarized as:

- We identify a general class of Gromov-Wasserstein problems, for point clouds embedded in low dimensional Euclidean spaces, that can be exactly represented as a concave low-rank QAP. In particular, mappings of images fits well within our framework.

- We develop a method for solving this class of Gromov-Wasserstein problems by solving a sequence of alternating sub-problems, which are either low-dimensional or linear.

- We prove that the proposed algorithm converges to a global optimal solution. The algorithm produces an optimality certificate in each iteration, in the form of upper- and lower bounds, which informs us of the potential suboptimality if the algorithm is terminated early.

- We present a numerical study, showing the efficiency of the proposed algorithm by comparing to other global optimization methods. We also illustrate the importance of globally solving Gromov-Wasserstein problems on a problem in computational biology.

In Section 2 we introduce the Gromov-Wasserstein problems and how it can be written as a QAP. In Section 3 we identify a class of Gromov-Wasserstein discrepancy problems that can be written as a concave relaxed QAPs problem, and in Section 4 we present the main methodology and an algorithm for solving this class of problems. Finally, in Section 5 we present numerical results and an application in computational biology.

## 2   The Gromov-Wasserstein discrepancy problem

Let $x_1 \ldots, x_n \in \mathcal{X}$ and $y_1 \ldots, y_n \in \mathcal{Y}$ be two sets of points and consider the problem of finding an assignment $\pi$ between the point sets such that the pairwise distances $d_{\mathcal{X}}(x_i, x_{i'})$ and $d_{\mathcal{Y}}(y_{\pi(i)}, y_{\pi(i')})$ are as close as possible for $i, i' = 1, \ldots, n$, where $d_{\mathcal{X}}$ and $d_{\mathcal{Y}}$ represents a notion of distance on the sets $\mathcal{X}$ and $\mathcal{Y}$, respectively. This can be formulated as the discrete Gromov-Wasserstein discrepancy

problem

$$\min_{\Gamma \in P} \frac{1}{2} \sum_{i,i',j,j'=1}^{n} (d_{\mathcal{X}}(x_i, x_{i'}) - d_{\mathcal{Y}}(y_j, y_{j'}))^2 \Gamma_{i,j} \Gamma_{i',j'}, \tag{1}$$

and where the assignment $\pi$ is represented by a permutation matrix $\Gamma$ and $P$ is the set of all $n \times n$ permutation matrices. The corresponding relaxed problem, where instead $\Gamma$ is in the set of doubly stochastic matrices, denoted by $\overline{P}$, is often referred to as the Gromov-Wasserstein problem [20]. In these formulations we note that

$$\sum_{i,i',j,j'=1}^{n} (d_{\mathcal{X}}(x_i, x_{i'}) - d_{\mathcal{Y}}(y_j, y_{j'}))^2 \Gamma_{i,j} \Gamma_{i',j'}$$

$$= \sum_{i,i',j,j'=1}^{n} (d_{\mathcal{X}}(x_i, x_{i'})^2 - 2d_{\mathcal{X}}(x_i, x_{i'})d_{\mathcal{Y}}(y_j, y_{j'}) + d_{\mathcal{Y}}(y_j, y_{j'})^2) \Gamma_{i,j} \Gamma_{i',j'}$$

$$= \langle C_x, C_x \rangle - 2\langle C_x \Gamma, \Gamma C_y \rangle + \langle C_y, C_y \rangle$$

where $C_x = [d_{\mathcal{X}}(x_i, x_{i'})]_{i,i'=1}^{n}$, $C_y = [d_{\mathcal{Y}}(y_j, y_{j'})]_{j,j'=1}^{n}$, and $\langle \cdot, \cdot \rangle$ denotes the standard (Frobenius) inner product. Since the first and third sums are independent of $\Gamma$, solving the discrete Gromov-Wasserstein problem (1) is the same as solving a quadratic assignment problem (QAP) on a simplified Koopmans-Beckmann form [5], namely as

$$\min_{\Gamma \in P} \quad -\langle C_x \Gamma, \Gamma C_y \rangle + \frac{1}{2}(\langle C_x, C_x \rangle + \langle C_y, C_y \rangle). \tag{2}$$

This problem is in general NP-hard, and the number of variables scales with the number of data points, making (2) computationally intractable for problems of relevant size. Here, we focus on instances where the matrices $C_x, C_y$ are positive definite and low rank. By utilizing this structure, we develop an algorithm that is guaranteed to find a globally optimal solution and scales well with the number of points.

## 3 The Gromov-Wasserstein problem and low rank QAP

An important special case of the Gromow-Wasserstein problem, considered in [20, 22], is when the point clouds belong to the Euclidean space and the squared Euclidean distance is used as discrepancy. That is, when the set of points are $x_1 \ldots, x_n \in \mathbb{R}^{\ell_x}$ and $y_1 \ldots, y_n \in \mathbb{R}^{\ell_y}$, which we represent by the matrices

$$X = (x_1, x_2, \ldots, x_n) \in \mathbb{R}^{\ell_x \times n}, \qquad Y = (y_1, y_2, \ldots, y_n) \in \mathbb{R}^{\ell_y \times n}.$$

In this case it can be noted that the distance matrices $C_x$ and $C_y$ has a rank bounded by $\ell_x + 2$ and $\ell_y + 2$, respectively, which can be seen from the identities

$$C_x = (\|x_i - x_j\|_2^2)_{i,j=1}^{n} = \mathbf{1} m_x^T - 2X^T X + m_x \mathbf{1}^T, \tag{3a}$$

$$C_y = (\|y_i - y_j\|_2^2)_{i,j=1}^{n} = \mathbf{1} m_y^T - 2Y^T Y + m_y \mathbf{1}^T, \tag{3b}$$

where $m_x = (\|x_1\|^2, \|x_2\|^2, \ldots, \|x_n\|^2)^T$, $m_y = (\|y_1\|^2, \|y_2\|^2, \ldots, \|y_n\|^2)^T$, and $\mathbf{1} \in \mathbb{R}^{n \times 1}$ is a column vector of ones. This observation was also used in [22] for formulating the Gromov-Wasserstein problem as a quadratic problem of rank $(\ell_x + 2)(\ell_y + 2)$ and developing fast algorithms for the problem. However, the rank can be even further reduced and the corresponding Gromov-Wasserstein problem can be formulated as a QAP problem of rank $\ell_x \ell_y$ [25, Lemma 4.2.3] (cf. [20, Proposition 1]).

**Proposition 1.** *[25, Lemma 4.2.3] Let $\Gamma$ be a doubly stochastic matrix and the matrices $C_x$ and $C_y$ given by* (3)*, then it holds that*

$$\langle C_x \Gamma, \Gamma C_y \rangle = \langle 2X\Gamma Y^T, 2X\Gamma Y^T \rangle + \langle L, \Gamma \rangle + 2\mathbf{1}^T m_y \mathbf{1}^T m_x,$$

*where $L = 2n m_x m_y^T - 4m_x \mathbf{1}^T Y^T Y - 4X^T X \mathbf{1} m_y^T$.*

*Proof.* The proposition follows by straightforward computations, where the expressions are simplified using $\mathbf{1} = \Gamma \mathbf{1} = \Gamma^T \mathbf{1}$. See the appendix for the proof. $\qquad \square$

Hence, the low rank QAP formulation of the discrete Gromov-Wasserstein problem can be stated as

$$\min_{\Gamma \in P} \quad -\langle 2X\Gamma Y^T, 2X\Gamma Y^T \rangle - \langle L, \Gamma \rangle + c_0 \tag{4}$$

where $L = 2nm_x m_y^T - 4m_x \mathbf{1}^T Y^T Y - 4X^T X \mathbf{1} m_y^T$, and $c_0 = (\langle C_x, C_x \rangle + \langle C_y, C_y \rangle - 4\mathbf{1}^T m_y \mathbf{1}^T m_x)/2$. The Gromov-Wasserstein problem can also be rewritten similarly and formulated as

$$\min_{\Gamma \in \overline{P}} \quad -\langle 2X\Gamma Y^T, 2X\Gamma Y^T \rangle - \langle L, \Gamma \rangle + c_0, \tag{5}$$

and since the objective function is concave, any optimal solution of (4) is also an optimal solution of the relaxed problem.

**Proposition 2.** *Any optimal solution of the discrete Gromov-Wasserstein problem* (4)*, is also an optimal solution to the Gromov-Wasserstein problem* (5)*. Conversely, problem* (5) *always has an optimal solution in one extreme point,[1] and any optimal extreme point to* (5) *is also an optimal solution to* (4)*.*

*Proof.* Since (5) is the minimization of a concave objective function over a convex sets $\overline{P}$, it attains the optimal value in an extreme point of the feasible set. Since the permutation matrices are the extreme points to the doubly stochastic matrices, i.e., $P = \text{ext}(\overline{P})$, (5) attains its minimum on $P$. Further, the set of points in $P$ for which (5) attains its minimum are the optimal solutions of (4). To show the converse statement, note that a minimum exists since $\overline{P}$ is compact and the objective function is continuous. Further, since the objective function is concave, an optimum must be at an extreme point. Finally, since the extreme points of $\overline{P}$ is the permutation matrices $P$, any optimal extreme point of (5) is also feasible and optimal to (4). $\qquad\square$

In the next section we will propose a methodology and an algorithm for solving this problem.

## 4 A cutting plane algorithm utilizing the low rank structure

By Proposition 2, we know that an optimal solution to the discrete Gromov-Wasserstein problem (4) can be obtained by solving the relaxed problem (5). However, the relaxed problem (5) is still a high-dimensional non-convex QP, which is NP-hard [19]. The high dimensionality can, in particular, be a limiting factor in solving the problem. For example, it is known that the performance of spatial branch-and-bound, one of the main approaches for globally optimizing nonconvex problems [10], can scale poorly with the number of variables. Thus, directly optimizing either (1) or (5) by spatial branch-and-bound is not computationally tractable for larger instances. Our idea is to use the low-rank formulation of the Gromov-Wasserstein problem and perform the optimization in a projected subspace of dimension $\ell_x \ell_y + 1$ by solving a sequence of relaxed problems.

First, we note that problem (5) can be written as

$$\min_{W \in \mathbb{R}^{\ell_x \times \ell_y}, w \in \mathbb{R}, \Gamma \in \overline{P}} \quad -\|W\|_F^2 - w + c_0 \tag{6a}$$

$$\text{subject to} \qquad W = 2X\Gamma Y^T, \ w = \langle L, \Gamma \rangle. \tag{6b}$$

Equivalence of problems (5) and (6) is shown by simply inserting the expressions for $w$ and $W$ into the objective function. Next, we project out the $\Gamma$ variables, and we define the feasible set in the $(w, W)$-space as $\mathcal{F} = \text{Proj}_{W,w} \left( W \in \mathbb{R}^{\ell_x \times \ell_y}, w \in \mathbb{R}, \Gamma \in \overline{P} \mid W = 2X\Gamma Y^T, \ w = \langle L, \Gamma \rangle \right)$.

Constructing an H-representation of the polytope $\mathcal{F}$, i.e., representing it by linear constraints of the form $\langle Z_r, W \rangle + \alpha_r w \leq \beta_r$, is not trivial and the number of constraints can grow exponentially with the number of data points. Therefore, we propose an algorithm based on a cutting plane scheme to optimize over $\mathcal{F}$.

---

[1]An extreme point of a convex set is a point in the set which does not lie in any open line segment joining two points of the set.

Instead of directly optimizing the objective in (6a) over the feasible set $\mathcal{F}$, which we don't have a tractable representation for, we relax the problem as

$$\min_{W \in \mathbb{R}^{\ell_x \times \ell_y}, w \in \mathbb{R}} \quad -\|W\|_F^2 - w + c_0 \tag{7a}$$

$$\text{subject to} \quad \langle Z_r, W \rangle + \alpha_r w \leq \beta_r, \quad \text{for } r = 1, \ldots, N. \tag{7b}$$

The linear constraints (7b) are supporting hyperplanes of the feasible set $\mathcal{F}$, which we will generate iteratively. The goal is to force the minimizer of problem (7) into the feasible set $\mathcal{F}$ by using relatively few linear constraints. Keep in mind, we don't need a full representation of set $\mathcal{F}$, we only need to capture the shape of $\mathcal{F}$ in some areas of interest, e.g., the constraints defining the faces of $\mathcal{F}$ at the optimal solution of problem (6) would suffice. The main advantage of the relaxation in problem (7) is that it contains far fewer variables than both problems (5) and (6), and the dimensionality is independent of the number of data points. Problem (7) can, therefore, be solved much more efficiently, especially in early iterations when the number of constraints is low. We will show that the constraints can be determined, as needed, by solving optimal transport problems. Based on this, we will develop an iterative approach that sequentially solves problem (7) and adds a constraint until the the solutions is the same as (6).

To initialize the search we determine a bounding box of $\mathcal{F}$ and use this to define a set of constraints (7b). The bounding box is determined by the (elementwise) minimum and maximum of the variables $w$ and $W$ given by

$$\min_{\Gamma \in \bar{P}} 2(X\Gamma Y^T)_{i,j} \leq W_{i,j} \leq \max_{\Gamma \in \bar{P}} 2(X\Gamma Y^T)_{i,j} \quad \text{for } i = 1, \ldots, \ell_x; \; j = 1, \ldots, \ell_y \tag{8a}$$

$$\min_{\Gamma \in \bar{P}} \langle L, \Gamma \rangle \leq \; w \; \leq \max_{\Gamma \in \bar{P}} \langle L, \Gamma \rangle, \tag{8b}$$

which can each be computed efficiently by solving a standard optimal transport problem. Initializing the set of constraints by the bounding box ensures that (7) is well-defined and bounded.

If the minimizer of problem (7) is within $\mathcal{F}$, then we can stop as the solution is optimal for (6).[2] Otherwise, we improve the outer approximation of $\mathcal{F}$ by adding new a constraint defined by $Z_{N+1}$, $\alpha_{N+1}$ and $\beta_{N+1}$. Let $(w_N, W_N)$ be the current optimal solution of (7), and assume that $(w_N, W_N) \notin \mathcal{F}$, then we form a new constraint, a so-called cutting plane, that excludes $(w_N, W_N)$ from the feasible set of (7).

We form a new constraint based on the gradient of the objective function (7a), which is given by

$$\nabla_{(w, \text{vec}(W)^T)}(-\|W\|_F^2 - w,) = \left(-1, -2\,\text{vec}(W)^T\right).$$

By letting $\alpha_{N+1} = 1$ and $Z_{N+1} = 2W_N$, the hyperplane defining the new constraint will have the (negative) gradient in the optimum $(w_N, W_N)$ as normal vector. Then we select $\beta_{N+1}$ such that the new constraint forms a supporting hyperplane of $\mathcal{F}$ (7b). This can be found by solving the following optimal transport problem

$$\beta_{N+1} := \max_{W \in \mathbb{R}^{\ell_x \times \ell_y}, w \in \mathbb{R}, \Gamma \in \overline{P}} \langle Z_{N+1}, W \rangle + \alpha_{N+1} w \qquad = \max_{\Gamma \in \overline{P}} \langle 4X^T W_N Y + L, \Gamma \rangle. \tag{9}$$

$$\text{subject to} \qquad W = 2X\Gamma Y^T, \; w = \langle L, \Gamma \rangle$$

When solving this problem, we also obtain a solution $\Gamma_N$ which is a doubly stochastic matrix (generically also a permutation matrix), which gives an upper bound for (6) and a candidate for the optimal solution. In the following subsection, we prove that the that algorithm, described in Algorithm 1, converges to a globally optimal solution.

A geometrical illustration of the algorithm is given in Figure (1). For illustrative purposes, we have used one-dimensional data resulting in a two-dimensional problem in the $(W, w)$-space. The data sets consist of 6 points each where one of the data sets has a reflective symmetry. This results in two global optima and 6! projected permutations in $P$. The first solution of (7) is located at one of the corners of the bounding box and marked with a "1" (the subsequent solutions are marked "2" – "5"). The infeasible point "1" is excluded from the search space by a cutting plane (red line, marked with an "A"). Following the same procedure we obtain point "2", and cutting plane "B". Adding further cutting planes excludes "3" and subsequently "4", resulting in the feasible and optimal point "5". Note that the cutting planes from iterations 3 and 4 almost overlap the cutting planes "A" and "B", since the gradients in the points 1 and 3 are very similar (the same for points 2 and 4).

---

[2]Remember, we are minimizing the objective (6a) over an outer approximation of the feasible set.

**Algorithm 1** Gromov-Wasserstein problem

---

**Input** $X \in \mathbb{R}^{\ell_x \times n}, Y \in \mathbb{R}^{\ell_y \times n}, \epsilon > 0$      (Define point clouds and give tolerance level)
$L_{\text{bound}} \leftarrow -\infty$, and $U_{\text{bound}} \leftarrow \infty$      (Set lower and upper bounds)
$(Z_r, \alpha_r, \beta_r)$ for $r = 1, \dots, N$ from (8), where $N = 2\ell_x \ell_y + 2$      (Set initial constraints)
**while** $U_{\text{bound}} - L_{\text{bound}} > \epsilon$ **do**
$(w_N, W_N) \leftarrow$ Optimal solution to (7)      (Solve (7))
$L_{\text{bound}} \leftarrow -\|W_N\|_F^2 - w_N + c_0$      (Update lower bound)
$\Gamma_N \leftarrow$ Optimal solution to (9)      (Solve (9))
$U_{\text{bound}} \leftarrow \min(U_{\text{bound}}, -\|2X\Gamma_N Y^T\|_F^2 - \langle L, \Gamma_N \rangle + c_0)$      (Update upper bound)
$(Z_{N+1}, \alpha_{N+1}, \beta_{N+1}) \leftarrow (2W_N, 1, \langle 4X^T W_N Y + L, \Gamma_N \rangle)$      (Calculate new constraints)
$N \leftarrow N + 1$      (Update iteration number)
**end while**

---

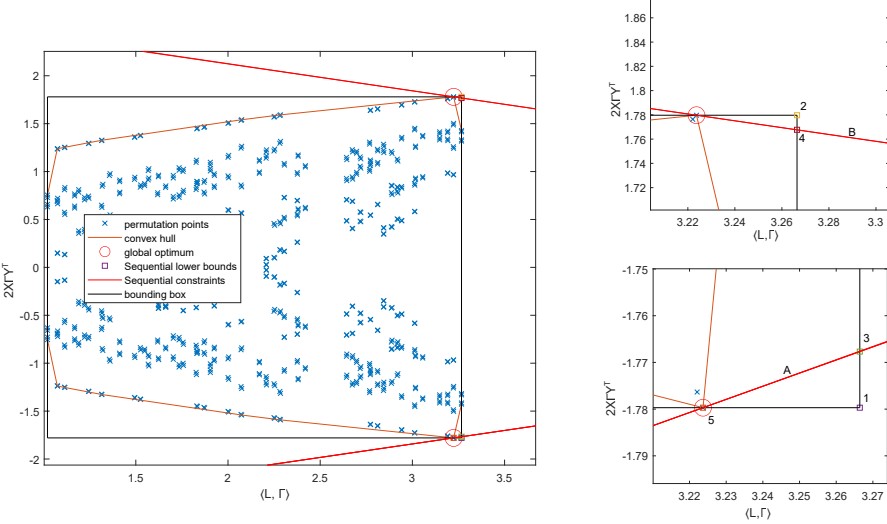

Figure 1: Left image: An illustrative example of the method on one-dimensional data. Right image: The area around the two global optima highlighting the sequence of optimal extreme points in the approximate cover and generation of cutting planes.

## 4.1 Proof of convergence of Algorithm 1

The main result considering convergence is presented in the following theorem.

**Theorem 1.** *The gap between the upper bound and lower bound in Algorithm 1 converges to $0$ (if the tolerance is $\epsilon = 0$).*

*Proof.* Consider the $N$th iteration in Algorithm (1), let $(w_N, W_N)$ be an optimal solution to (7), and $\Gamma_N$ is an optimal solution to (9) with corresponding points $(\hat{w}, \hat{W}) = (\langle L, \Gamma_N \rangle, 2X\hat{\Gamma}_N Y^T)$, in the $(w, W)$-space. Assume that the gap in the objective function between those two points is

$$\epsilon_N = \|W_N\|_F^2 + w_N - \|\hat{W}\|_F^2 - \hat{w}. \tag{10}$$

The new constraint is then defined by $\langle Z_{N+1}, W \rangle + w \leq \beta_{N+1}$ where $Z_{N+1} = 2W_N$ and $\beta_{N+1} = 2\langle W_N, \hat{W} \rangle + \hat{w}$, and thus for any point $(w, W)$ that satisfy the constraint it must hold that

$$0 \leq 2\langle W_N, \hat{W} - W \rangle + \hat{w} - w.$$

By substituting $\hat{w}$ from (10), we obtain

$$
\begin{aligned}
\epsilon_N &\leq 2\langle W_N, \hat{W} - W\rangle - w + \|W_N\|_F^2 + w_N - \|\hat{W}\|_F^2 \\
&= w_N - w + 2\langle W_N, W_N - W\rangle - \|\hat{W} - W_N\|_F^2 \\
&\leq w_N - w + 2\langle W_N, W_N - W\rangle \\
&\leq (|w_N - w|^2 + \|W_N - W\|_F^2)^{1/2}(1 + 4\|W_N\|_F^2)^{1/2}
\end{aligned}
$$

where we in the last step have used the Cauchy-Schwarz inequality.

For any iteration number $N+k$ with $k > 0$, we have that $(w_{N+k}, W_{N+k})$ is feasible for $\langle Z_{N+1}, W\rangle + w \leq \beta_{N+1}$, and thus the Euclidean distance between $(w_N, W_N)$ and $(w_{N+k}, W_{N+k})$ is at least $\epsilon_N/(1 + 4\|W_N\|_F^2)$. If the gap in the algorithm does not converge to 0, then there is an $\epsilon > 0$ for which $\epsilon_N \geq \epsilon$ for all $N$ and thus the distance between any two points in the sequence $\{(w_N, W_N)\}_N$ is bounded from below by $\epsilon/(1 + 4\max\{\|W\|_F^2 \mid W \in (8)\})$. However, since the infinite sequence of points $\{(w_N, W_N)\}_N$ belong to a bounded set defined by (8), there must be a convergent subsequence, which contradicts that there is a positive lower bound on the distance between any two points. $\qquad\square$

From Theorem 1 the gap between the upper and lower bound converges to zero, and thus Algorithm 1 converges to a globally optimal solution.

### 4.2 Considerations when solving the relaxed problem

Problem (7) minimizes a concave function over a convex set. Thus, the solution is located in the extreme points of the convex set, i.e., the outer approximation of $\mathcal{F}$. The standard approach to solve such problems is by branch and bound methods. However, the low dimension and sequential generation of constraints make it viable to search among the extreme points for an optimal solution.

To simplify notation, we define $x^T = \begin{pmatrix} w & \text{vec}(W)^T \end{pmatrix} \in \mathbb{R}^r$ where $r := \ell_x \ell_y + 1$. Then we can write (7b) on the form $A_k x \leq b_k$. Note that, by construction, none of the constraints are strongly redundant as every constraint is satisfied with equality for a permutation. As the constraints are added sequentially, it is actually easy to compute the new extreme points by keeping track of previous extreme points as described in the following proposition.

**Proposition 3.** *Assume that the extreme points $\{x_k\}_k$ of the convex set described by $Ax \leq b$ are known. When adding a constraint $A_N^T x \leq b_N$, the additional extreme points are linear combinations of pairs of existing extreme points $x_{k_1}$ and $x_{k_2}$ both satisfying the same $r-1$ constraints with equality and $A_N^T x_{k_1} \leq b_n$ and $A_N^T x_{k_2} > b_n$ so that the combination satisfies $A_N^T(\lambda x_{k_1} + (1-\lambda)x_{k_2}) = b_N$.*

*Proof.* This is done by counting the number of constraints satisfied with equality. See the appendix for the proof. $\qquad\square$

Especially, in lower dimensions, e.g., with two or three dimensional data, this approach of keeping track of all extreme points and calculating new extreme points after adding a constraint can be very efficient for solving problem (7). More details of this method is provided in the appendix. In the numerical results, we present results where problem (7) is solved both by this extreme point search and the spatial branch and bound method in Gurobi.

## 5 Numerical results

### 5.1 Computational efficiency

In this section we compare the time to solve the problem up to an accuracy measured in relative error with different methods: Algorithm 1 when (7) is solved with the extreme point method as described in section 4.2, Algorithm 1 when (7) solved using Branch & bound using Gurobi 10.0 [13], MILP1 formulation in [11] implemented in Gurobi and finally when (6) is directly solved using Gurobi. The MILP1 formulation can handle a larger class of problems, but is reported to handle very few dimensions. All computations were performed using Matlab on an Intel i5 2.9 GHz PC. The linear optimal mass problem (9) was solved using the package [3] which is based on the network simplex [18]. The model problems tested are evenly distributed points in a unit disc or ball which we denote

Table 1: Computational efficiency. Computational time on the format [mean (low - high)] from 5 repeats for various problem geometries, dimensions and sizes, for the proposed extreme point method, the same method using branch and bound (B&B), the problem formulation MILP1 [11] and finally (6) implemented in Gurobi via Matlab interface. An "-" indicates that the problem timed out, in such a way being incomparable with the proposed method. An "!" indicates that the problem reached $10^4$ iterations and stopped to the accuracy indicated.

| Type | $n$ | $\ell_x,\ell_y$ | Rel. error | Algorithm 1 [s] Extreme point / B&B | MILP1 [s] | (6) B&B [s] |
|---|---|---|---|---|---|---|
| $\mathcal{U}$ | 10 | 2,2 | $10^{-8}$ | 0.14 (0.07-0.3) / 21 (6-47) | 39 (11-58) | 0.15 (0.14-0.16) |
| $\mathcal{U}$ | 100 | 2,2 | $10^{-8}$ | 0.48 (0.3-0.7) / 86 (52-107) | - | 25 (19-39) |
| $\mathcal{U}$ | 500 | 2,2 | $10^{-8}$ | 11 (9-16) / 408 (269-511) | - | - |
| $\mathcal{U}$ | 1000 | 2,2 | $10^{-8}$ | 69 (54-85) / 576 (389-1059) | - | - |
| $\mathcal{U}$ | 2000 | 2,2 | $10^{-8}$ | 460 (313-653) / - | - | - |
| $\mathcal{U}$ | 10 | 2,3 | $10^{-8}$ | 1.8 (1.2-2.4) / 133 (45-296) | 105 (49-147) | 2.4(1.8-3.4) |
| $\mathcal{U}$ | 100 | 2,3 | $10^{-8}$ | 278 (99-813) / - | - | 172 (133-221) |
| $\mathcal{U}$ | 500 | 2,3 | $10^{-8}$ | 9568 / - | - | - |
| $\mathcal{N}_1$ | 10 | 2,3 | $10^{-8}$ | 0.51 (0.39-0.65) / 708 (233-1184) | 146 (66-227) | 3 (2.6-4.0) |
| $\mathcal{N}_1$ | 100 | 2,3 | $10^{-8}$ | 86 (20-275) / - | - | 95 (73-116) |
| $\mathcal{N}_1$ | 500 | 2,3 | $10^{-5}$ | 5310!/ - | - | - |
| $\mathcal{N}_2$ | 10 | 3,3 | $10^{-2}$ | 1.8 (0.7-3.2) / 142 (73-210) | 117 (71-163) | 0.2(0.1-0.3) |
| $\mathcal{N}_2$ | 100 | 3,3 | $10^{-2}$ | 36 (22-55)/ - | - | 45(36-65) |
| $\mathcal{N}_2$ | 500 | 3,3 | $10^{-2}$ | 436 (228-862) / - | - | - |
| $\mathcal{N}_3$ | 10 | 3,3 | $10^{-2}$ | 1.2 (0.5-2.3) / 22 (11-43) | 72 (43-94) | 0.2(0.1-0.3) |
| $\mathcal{N}_3$ | 100 | 3,3 | $10^{-2}$ | 7 (5-8)/ 91 (76-111) | - | 10 (9-12) |
| $\mathcal{N}_3$ | 500 | 3,3 | $10^{-2}$ | 11 (9-16) / 161 (104-226) | - | - |
| $\mathcal{N}_3$ | 1000 | 3,3 | $10^{-2}$ | 25 (22-29) / 176 (149-224) | - | - |
| $\mathcal{N}_3$ | 2000 | 3,3 | $10^{-2}$ | 93 (91-100) / 578 (429-691) | - | - |

$\mathcal{U}$, and normally distributed points $\mathcal{N}(0,\sigma)$. We denote $\mathcal{N}_1 := \mathcal{N}(0, I)$, $\mathcal{N}_2 := \mathcal{N}(0, \mathrm{diag}(1, 1, \frac{1}{10}))$ and $\mathcal{N}_3 := \mathcal{N}(0, \mathrm{diag}(1, \frac{1}{2}, \frac{1}{10}))$. See Table 1 for numerical results. Some notes on the results

1. On 2-dimensional data ($\ell_x = \ell_y = 2$), the extreme point method is particularly efficient.

2. For problems that need many extreme points ($> 10^6$), which depends on the data itself, the handling of extreme points becomes the driver of computational cost.

3. Problems mainly containing reflections (e.g. $\mathcal{N}_3$) are easier to solve than those with room for rotations.

4. Directly solving (6) with Gurobi was not feasible for problems with $n \geq 500$.

## 5.2 Comparison with local search method

We compare the results using the proposed method to a local search method provided in the Git-hub repository for [20]. The entropy regularization parameter is set to 0, and the method is run with random initializations (including the first lower bound [17] used with success in [22]) until the relative error to the global optimum is less than a specific tolerance $\epsilon$. The problems are the same as in the previous section. Note here that in the local search methods we need an oracle in order to determine when we have reached a given performance level (which of course is not available in practice), whereas we in the proposed method computes upper and lower bounds.

Results for two and three dimensional data ($\ell_x = \ell_y = 2, 3$) are presented in Table 2. The results show that the proposed method performs better than multi-starting the local method when $\ell_x = \ell_y = 2$. For the matching of 2-dimensional data to 3-dimensional data, the local search method is surprisingly fast suggesting that the problems is of a completely different nature than when 2-d data is matched to 2-d data or 3-d data is matched to 3-d data.

Note that we have chosen to compare with the method from [20] rather than [22]. This is since we have not optimized the optimal transport computations using the low rank structures in the problem. If we optimized the computations in this way, i.e., as in [21], we expect to get similar improvement as in [22] compared with [20].

Table 2: Computational efficiency compared with local search [20]. Computational time on the format [mean (low - high)] on two specific problems which contain near symmetries and the number of random initializations needed to achieve the required accuracy on the format [mean (low - high)]. The number of initializations were limited to 1000.

| Type | $n$ | $\ell_x, \ell_y$ | Rel. error $\epsilon$ | Algorithm 1 Exec. time [s] | Local search Exec. time [s] | Local search Initializations | Sucessful runs |
|------|-----|-------|-----------------|-----------------|-----------------|-----------------|-----------------|
| $\mathcal{U}$ | 100 | 2,2 | $10^{-6}$ | 0.5 (0.4-0.6) | 4 (0.3-12.7) | 64 (4-205) | 5 |
| $\mathcal{U}$ | 200 | 2,2 | $10^{-6}$ | 1.6 (1.0-1.9) | 27 (13-42) | 129 (59-197) | 5 |
| $\mathcal{U}$ | 300 | 2,2 | $10^{-6}$ | 3.2 (2.7-3.9) | 174 (33-458) | 366 (69-959) | 5 |
| $\mathcal{U}$ | 400 | 2,2 | $10^{-6}$ | 6 (5-8) | 322 (97-685) | 321 (97-781) | 3 |
| $\mathcal{U}$ | 100 | 2,3 | $10^{-6}$ | 352 (99-669) | 23 (8-60) | 341 (110-853) | 5 |
| $\mathcal{U}$ | 200 | 2,3 | $10^{-6}$ | 1238 (643-2612) | 92 (11-168) | 421(49-778) | 3 |
| $\mathcal{U}$ | 300 | 2,3 | $10^{-6}$ | 3006 (601-4908) | 131 (4-344) | 270 (10-710) | 5 |
| $\mathcal{U}$ | 400 | 2,3 | $10^{-6}$ | 4729 (4279-4868) | 367 (5-343) | 365 (1-859) | 5 |
| $\mathcal{N}_1$ | 100 | 2,2 | $10^{-6}$ | 0.5 (0.3-0.7) | 0.6 (0.2-1.0) | 10 (3-16) | 5 |
| $\mathcal{N}_1$ | 200 | 2,2 | $10^{-6}$ | 0.9 (0.7-1.1) | 13.4 (1-37) | 61 (6-168) | 5 |
| $\mathcal{N}_1$ | 300 | 2,2 | $10^{-6}$ | 2.4 (2.0-2.9) | 37.5 (0.5-90) | 75 (1-182) | 5 |
| $\mathcal{N}_1$ | 400 | 2,2 | $10^{-6}$ | 4.4 (3.6-5.3) | 149 (22-378) | 142 (21-361) | 4 |

## 5.3 Application to symmetrical data for morphological analysis

In this example we investigate the impact of correctly evaluating the Gromov-Wasserstein discrepancy compared to estimating it by local search. As a test case, we examine the ability to classify Adeno Associated Viral (AAV) particles based on the Gromov-Wasserstein discrepancy on image data originating from transmission electron microscopy, hence $\ell_x = \ell_y = 2$. AAV particles are nearly round viral particles with multiple near-rotational symmetries as illustrated in Figure 2. By sampling $n = 500$ positions on each AAV particle proportional to the protein density, the point sets from pairs of particles $X_i$ and $X_j$ can subsequently be compared using the Gromov-Wasserstein discrepancy.

Computing the Gromov-Wasserstein discrepancy between all objects in a large set $\mathcal{X} = \{X_i\}_i^N$ is tedious. Therefore, one may consider calculating the discrepancy to a subset of the objects that are well distributed under the Gromov-Wasserstein discrepancy. To find such a subset without actually calculating all pairs of discrepancies, we use a greedy approach by defining the index subset $S_k = \{s_i\}_{i=1}^k$ by selecting the first object arbitrarily and then let the set grow by

$$S_{k+1} := \{S_k, \operatorname*{argmax}_{i \notin S_k} \min_{j \in S_k} d_{GW}(X_i, X_j)\}. \tag{11}$$

In this way all objects are closer than a tolerance to an object in the subset, and the way the subset is produced generates a monotonically decreasing tolerance. By using this procedure, every object obtains a feature vector of distances to the objects indexed by $S_k$. Next the feature vectors are used as input to a K-means clustering and classification quality in terms of purity, adjusted rand index and normalized mutual information compared to an expert evaluation is presented in Figure 3.

The example shows that if the data contains symmetries, local search methods may get stuck on permutations that are locally optimal but that are far from globally optimal, see Figure 2. Thus the discrepancy obtained from local search is inappropriate to use as a discriminating feature, as shown in the lower performance in the classification illustrated in Figure 3. In the left subfigure of Figure 2, self-similarities are visited in the proposed method, and these local optima are in fact also almost optimal when used as initiation point using local search methods e.g., [20].

This example shows that when the Gromov-Wasserstein problem is calculated accurately it provides valuable information and biological meaning as it differentiates viral particles with different cargo and

variations in capsid structure, and, at the same time, finds the optimal orientation positively revealing possible chiralities or isomerisms. It also shows that when the distance is calculated accurately, it provides better decision support than using local search methods.

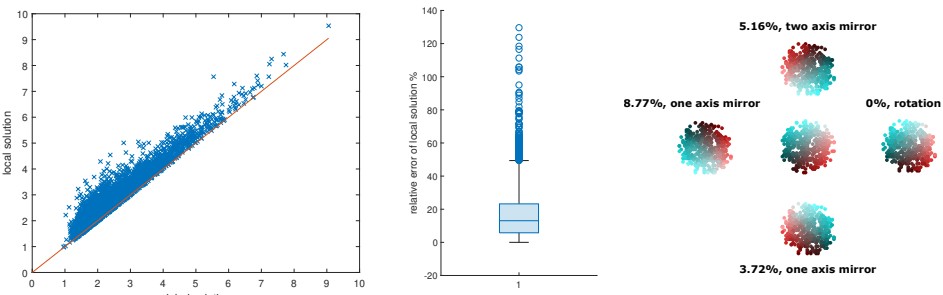

Figure 2: Left: Each distance calculated with the proposed method and the same distance calculated with the local search method [20]. For near symmetrical data, the confusion of the measurements of the local method is clear. Middle: The relative error of the local method compared to the global solution. Right: Matching of the structure to itself to four different orientations visited by the algorithm where the color indicate the permutation $\Gamma$. The relative error of the Gromov-Wasserstein-discrepancy and the isometry to the global optimum is written near each matching. The global optimum is able to correctly match to itself, whereas the local search method may get stuck in local optima.

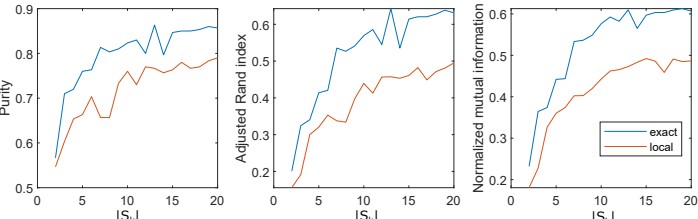

Figure 3: The trajectory of the increased quality of classification compared to an expert evaluation when the distance is computed from all particles to the sequence of particles suggested in the text. The gain of quality using an exact evaluation (blue) of the Gromov-Wasserstein problem is unambiguous over the local search (red).

## 6 Discussion

When using distances as input for statistical analyses, the accuracy of the measurement set a bound for the information resolution. If the measurement system introduces error of a certain structure, this can produce artefacts in the result and affect decisions taken on the result. When using distances for such purposes, it is necessary to either know the measurement error, the artefacts being produced, or using an accurate measurement system. In this paper we have provided a method which computes the Gromov-Wasserstein problem accurately, which reduces the uncertainty of such considerations.

## Acknowledgement

This work was funded by Vironova AB and the Swedish innovation agency Vinnova through the innovation milieu GeneNova (2021-02640).

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

# A  Appendix

## A.1  Proof of Proposition 1

**Proposition 4.** *Let $\Gamma$ be a doubly stochastic matrix and the matrices $C_x$ and $C_y$ given by*

$$C_x = (\|x_i - x_j\|_2^2)_{i,j=1}^n = \mathbf{1}m_x^T - 2X^TX + m_x\mathbf{1}^T,$$
$$C_y = (\|y_i - y_j\|_2^2)_{i,j=1}^n = \mathbf{1}m_y^T - 2Y^TY + m_y\mathbf{1}^T,$$

*then it holds that*

$$\langle C_x\Gamma, \Gamma C_y \rangle = \langle 2X\Gamma Y^T, 2X\Gamma Y^T \rangle + \langle L, \Gamma \rangle + 2\mathbf{1}^T m_y \mathbf{1}^T m_x,$$

*where $L = 2nm_x m_y^T - 4m_x\mathbf{1}^T Y^T Y - 4X^T X\mathbf{1}m_y^T$.*

*Proof.* The proposition follows by the following straightforward computations, where we just expand the expressions and use $\mathbf{1} = \Gamma\mathbf{1} = \Gamma^T\mathbf{1}$, thus

$$
\begin{aligned}
\operatorname{tr}(C_x\Gamma C_y\Gamma^T) =\ & \operatorname{tr}((\mathbf{1}m_x^T - 2X^TX + m_x\mathbf{1}^T)\Gamma(\mathbf{1}m_y^T - 2Y^TY + m_y\mathbf{1}^T)\Gamma^T) \\
=\ & m_x^T\Gamma(\mathbf{1}m_y^T - 2Y^TY + m_y\mathbf{1}^T)\Gamma^T\mathbf{1} \\
& - 2\operatorname{tr}(X^TX\Gamma(\mathbf{1}m_y^T - 2Y^TY + m_y\mathbf{1}^T)\Gamma^T) \\
& + \mathbf{1}^T\Gamma(\mathbf{1}m_y^T - 2Y^TY + m_y\mathbf{1}^T)\Gamma^T m_x \\
=\ & m_x^T\mathbf{1}m_y^T\mathbf{1} - 2m_x^T\Gamma Y^TY\mathbf{1} + nm_x^T\Gamma m_y \\
& - 2m_y^T\Gamma^TX^TX\mathbf{1} + 4\operatorname{tr}(X^TX\Gamma Y^TY\Gamma^T) - 2\mathbf{1}^TX^TX\Gamma m_y \\
& + nm_y^T\Gamma^T m_x - 2\mathbf{1}^TY^TY\Gamma^T m_x + \mathbf{1}^T m_y\mathbf{1}^T m_x \\
=\ & 4\operatorname{tr}(X^TX\Gamma Y^TY\Gamma^T) - 4m_x^T\Gamma Y^TY\mathbf{1} + 2nm_x^T\Gamma m_y \\
& - 4\mathbf{1}^TX^TX\Gamma m_y + 2\mathbf{1}^T m_y\mathbf{1}^T m_x \\
=\ & \langle 2X\Gamma Y^T, 2X\Gamma Y^T \rangle \\
& + \langle 2nm_x m_y^T - 4m_x\mathbf{1}^TY^TY - 4X^TX\mathbf{1}m_y^T, \Gamma \rangle \\
& + 2\mathbf{1}^T m_y\mathbf{1}^T m_x.
\end{aligned}
$$

$\square$

## A.2  Proof of Proposition 3

**Proposition 6.** *Assume that the extreme points $\{x_k\}_k$ of the convex set described by $Ax \leq b$ are known. When adding a constraint $A_N^T x \leq b_N$, the additional extreme points are linear combinations of pairs of existing extreme points $x_{k_1}$ and $x_{k_2}$ both satisfying the same $r-1$ constraints with equality and $A_N^T x_{k_1} \leq b_n$ and $A_N^T x_{k_2} \geq b_n$ so that the combination satisfies $A_N^T(\lambda x_{k_1} + (1-\lambda)x_{k_2}) = b_N$.*

*Proof.* Let $W$ be a matrix whose columns consist of the extreme points defined by the $N-1$ constraints $Ax \leq b$. Also, let $a_N^T x \leq b_N$ be an additional constraint, $e_k$ be a unit vector with 1 on position $k$, and let $\alpha$ parametrize the convex cone on $W$, i.e. $\mathbf{1}^T\alpha = 1$, $\alpha \geq 0$ so that $AW\alpha \leq b$ describes all points in the convex set. Suppose that $B_k$ describes the indices of the constraints that define the k:th extreme point by letting $A_{B_k}$ be the sub matrix of $A$ including the rows denoted by the indices in $B_k$. Then $A_{B_k}We_k = b_{B_k}$. It then follows that $AW(\lambda e_{k_1} + (1-\lambda)e_{k_2})_j = b_j$ if and only if $j \in B_{k_1}$ and $j \in B_{k_2}$ and $0 \leq \lambda \leq 1$. The construction in the proposition, $\lambda x_{k_1} + (1-\lambda)x_{k_2}$ so that $A_N^T(\lambda x_{k_1} + (1-\lambda)x_{k_2}) = b_N$ then satisfies $r$ constraints with equality and all other constraints with inequality, i.e. the point is an extreme point to the set. We also note that every multiple combination (three or more) of extreme points sharing $r-1$ constraints are not extreme points as they are linear combination of the pairs given by $\alpha(\lambda_1 x_1 + (1-\lambda_1)x_2) + (1-\alpha)(\lambda_2 x_2 + (1-\lambda_2)x_3)$. Every linear combination of pairs of points sharing less than $r-1$ constraints will not satisfy $r$ constraints, i.e. they are not extreme points. $\square$

## A.3 A description of the implementation of the extreme point method

The handling of the extreme points in the paper is done by keeping track of the extreme points, their connection to the boundary constraints, and lookup tables for the adjacent extreme points, i.e., extreme points that satisfies the same $r-1$ constraints with equality, where $r$ is the rank of the problem. Let the extreme points be described in the matrix $E$ where each column describes an extreme point. Thus $AE \leq b\mathbf{1}$, if the constraints are described by the matrix $A$ and vector $b$ such that $Ax \leq b$ is the constraint equations.

Let the adjacency be described in a (sparse) matrix with binary elements $D$ where $D_{i,j} = 1$ if extreme point $i$ and $j$ are adjacent. Let us also keep track of the constraints that are satisfied by an extreme point with equality. For this purpose let $B$ be a matrix with binary elements in which the element $B_{i,j} = 1$ if extreme point $j$ satisfies constraint $i$ with equality. Thus, $D_{i,j} = 1$ if $(B^T B)_{i,j} = r - 1$. This is one of the computational drivers for the proposed extreme point method.

When a new constraint $(A_n, b_n)$ is added, the new extreme points are generated by a linear combination of the infeasible extreme points $\mathcal{I} := \{i : A_n E_i > b_n\}$ and their adjacent feasible extreme points $f_i := \{j : D(i,j) = 1\}$ where $i \in \mathcal{I}$. Let $\#$ indicate the number of elements of a finite set, then $\sum_{i \in \mathcal{I}} \#(f_i)$ is the number of new extreme points. We place the new extreme points in the matrix $E$ by adding them in the end as a matrix $P$. The new matrix containing the extreme points

$$E_n = (E \quad P)$$

The matrix keeping track of which extreme points satisfies which constraints with equality is extended with

$$B_n = \begin{pmatrix} B & C \\ 0 & \mathbf{1} \end{pmatrix}$$

where $C_{\cdot,k} = B_{\cdot,i} \odot B_{\cdot,(f_i)_k}$, preferably implemented using bitwise operators. Here, the last row describes the newly added constraint and $k$ a re-enumeration of the new extreme points.

The new adjacency matrix $D_n$ can be concatenated with the old $D$ and two additional matrices

$$D_n = \begin{pmatrix} D & O \\ O^T & N \end{pmatrix}$$

where $O_{i,j} = 1$ the old extreme point $i$ is adjacent to the new extreme point $j$. This information is already available for us, since the new extreme points are adjacent to $f_j$. Finally, $N_{i,j} = 1$ if $(C^T C)_{i,j} = r - 2$. This is by far the most computationally expensive operation in the proposed algorithm, which can be implemented with std::popcount in the standard c++ library. For 3-dimensional problems, around 120 bits needs to be compared between all new extreme points.

## A.4 Additional tests

## A.5 Convergence rate

The convergence proof in Proposition 3 does not include a rate of convergence. In Figure 4, we show the convergence trajectory for the problems $\mathcal{U}$ and $\mathcal{N}_1$ for 2-dimensional data. The tests show that the convergence rate is linear to its nature up to a number of iterations where the gap closes completely.

In comparison to the result of the proposed method we present the same convergence evaluation with the local search method in [20]. Figure5 shows the trajectory of convergence. The tests show that the rate of convergence is sublinear to its nature. The number of initializations needed to achieve a pre-determined accuracy increase with the number of points for the local search. Note here that in order to determine when to stop one needs to know the optimal value, which is not available for the local search method.

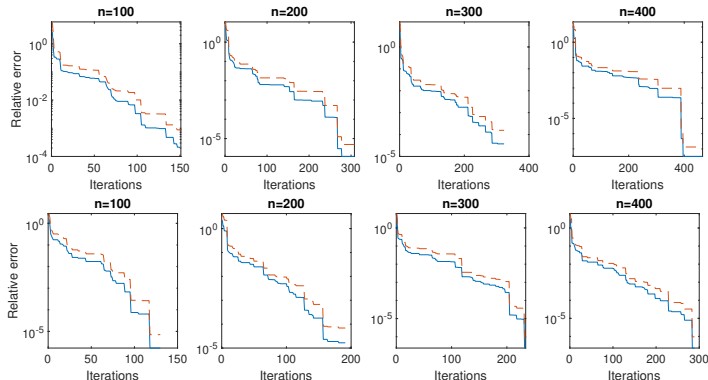

Figure 4: Top row: The trajectory of the relative error for the proposed method for the problem $\mathcal{U}$ on 2-dimensional data. Bottom row: The trajectory of the relative error for the proposed method for the problem $\mathcal{N}_1$ on 2-dimensional data. The solid blue line indicates the mean convergence rate for 20 runs and the dashed red line indicates 1 standard deviation from the mean.

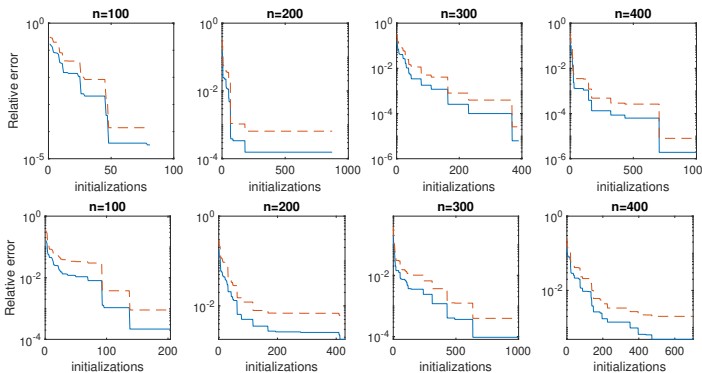

Figure 5: Top row: The trajectory of the relative error for the local search method for the problem $\mathcal{U}$ on 2-dimensional data. Bottom row: The trajectory of the relative error for the local search method for the problem $\mathcal{N}_1$ on 2-dimensional data. The solid blue line indicates the mean convergence rate for 20 runs and the dashed red line indicates 1 standard deviation from the mean.

## A.6 Evaluation on the MNIST dataset

In this section we present the performance of the proposed method on shapes originating from the MNIST dataset [8]. Even though the arabic numerals are not entirely unique to reflections (e.g. "2" and "5") and rotations (e.g. "6" and "9"), the ability to compute the exact Gromov-Wasserstein discepancy is of interest. When the images doesn't contain the same number of points, evenly distributed points are taken from the image with more points. The maximum number of points was set to 400. Number of iterations are presented in Table 3 and the computational time is presented in Table 4, showing that the proposed method works as anticipated on the data set. Examples of correspondances of numerals are shown in Figure 6.

Table 3: Number of iterations to compute the distance between two numeral shapes to a relative error gap of $10^{-8}$. Numbers presented are median in the upper table and [min - max] in the lower table. The number of random tests were 10 for each combination.

|   | 0 | 1 | 2 | 3 | 4 | 5 | 6 | 7 | 8 | 9 |
|---|---|---|---|---|---|---|---|---|---|---|
| 0 | 92.5 | 55.5 | 98.5 | 75.5 | 115.5 | 106.5 | 80 | 81 | 90 | 90 |
| 1 |   | 21 | 46.5 | 36 | 70.5 | 50.5 | 47 | 41 | 50 | 49 |
| 2 |   |   | 52 | 53.5 | 64 | 58.5 | 60.5 | 57.5 | 58 | 43 |
| 3 |   |   |   | 40.5 | 75 | 52.5 | 72.5 | 46.5 | 56 | 56.5 |
| 4 |   |   |   |   | 84.5 | 57 | 63.5 | 65 | 73.5 | 61.5 |
| 5 |   |   |   |   |   | 50 | 53.5 | 48 | 64.5 | 49.5 |
| 6 |   |   |   |   |   |   | 44 | 45.5 | 67 | 56.5 |
| 7 |   |   |   |   |   |   |   | 48.5 | 64 | 47.5 |
| 8 |   |   |   |   |   |   |   |   | 62.5 | 55.5 |
| 9 |   |   |   |   |   |   |   |   |   | 52.5 |

|   | 0 | 1 | 2 | 3 | 4 | 5 | 6 | 7 | 8 | 9 |
|---|---|---|---|---|---|---|---|---|---|---|
| 0 | 67-134 | 39-80 | 67-223 | 51-139 | 86-184 | 69-131 | 62-97 | 63-143 | 73-102 | 53-163 |
| 1 |   | 10-39 | 32-78 | 30-64 | 49-98 | 36-61 | 35-88 | 27-69 | 43-59 | 27-93 |
| 2 |   |   | 29-77 | 36-78 | 37-75 | 44-88 | 47-95 | 38-72 | 43-86 | 36-83 |
| 3 |   |   |   | 29-67 | 31-116 | 47-60 | 42-89 | 27-77 | 48-85 | 27-87 |
| 4 |   |   |   |   | 50-107 | 40-111 | 40-111 | 52-92 | 47-108 | 50-80 |
| 5 |   |   |   |   |   | 31-74 | 45-88 | 43-87 | 54-94 | 36-97 |
| 6 |   |   |   |   |   |   | 26-80 | 36-88 | 54-123 | 32-94 |
| 7 |   |   |   |   |   |   |   | 24-76 | 52-75 | 30-87 |
| 8 |   |   |   |   |   |   |   |   | 40-82 | 42-71 |
| 9 |   |   |   |   |   |   |   |   |   | 29-65 |

Table 4: Computational time for the distance between two numeral shapes to a relative error gap of $10^{-8}$. Numbers presented are [median]s in the upper table and [min - max]s in the lower table. The number of random tests were 10 for each combination.

|   | 0 | 1 | 2 | 3 | 4 | 5 | 6 | 7 | 8 | 9 |
|---|---|---|---|---|---|---|---|---|---|---|
| 0 | 0.85 | 0.48 | 1.03 | 0.76 | 1.09 | 1.33 | 0.83 | 0.81 | 0.98 | 0.95 |
| 1 |   | 0.05 | 0.18 | 0.15 | 0.32 | 0.22 | 0.18 | 0.15 | 0.3 | 0.29 |
| 2 |   |   | 0.34 | 0.33 | 0.41 | 0.39 | 0.46 | 0.37 | 0.47 | 0.34 |
| 3 |   |   |   | 0.31 | 0.7 | 0.41 | 0.64 | 0.39 | 0.52 | 0.51 |
| 4 |   |   |   |   | 0.85 | 0.61 | 0.6 | 0.67 | 0.86 | 0.7 |
| 5 |   |   |   |   |   | 0.53 | 0.62 | 0.49 | 0.75 | 0.5 |
| 6 |   |   |   |   |   |   | 0.54 | 0.49 | 0.96 | 0.66 |
| 7 |   |   |   |   |   |   |   | 0.61 | 0.84 | 0.59 |
| 8 |   |   |   |   |   |   |   |   | 0.77 | 0.74 |
| 9 |   |   |   |   |   |   |   |   |   | 0.92 |

|   | 0 | 1 | 2 | 3 | 4 | 5 | 6 | 7 | 8 | 9 |
|---|---|---|---|---|---|---|---|---|---|---|
| 0 | 0.5-1.4 | 0.3-1 | 0.6-2.4 | 0.5-1.5 | 0.9-1.9 | 0.6-1.6 | 0.6-1.2 | 0.5-1.7 | 0.6-1.4 | 0.5-2.1 |
| 1 |   | 0-0.1 | 0.1-0.4 | 0.1-0.4 | 0.2-0.4 | 0.2-0.3 | 0.1-0.4 | 0.1-0.4 | 0.2-0.4 | 0.1-0.5 |
| 2 |   |   | 0.2-0.5 | 0.2-0.6 | 0.2-0.5 | 0.3-0.6 | 0.3-0.8 | 0.3-0.5 | 0.3-0.7 | 0.2-0.7 |
| 3 |   |   |   | 0.2-0.7 | 0.2-1.2 | 0.3-0.5 | 0.3-0.9 | 0.2-0.7 | 0.4-0.7 | 0.2-0.8 |
| 4 |   |   |   |   | 0.4-1 | 0.3-1 | 0.3-1.3 | 0.4-0.8 | 0.4-1.3 | 0.4-0.9 |
| 5 |   |   |   |   |   | 0.2-0.9 | 0.4-1 | 0.3-0.9 | 0.5-1.1 | 0.4-1.1 |
| 6 |   |   |   |   |   |   | 0.2-1 | 0.3-1 | 0.6-1.8 | 0.3-1.2 |
| 7 |   |   |   |   |   |   |   | 0.3-0.9 | 0.7-1.2 | 0.1-1.4 |
| 8 |   |   |   |   |   |   |   |   | 0.6-1.4 | 0.5-1.1 |
| 9 |   |   |   |   |   |   |   |   |   | 0.5-1.1 |

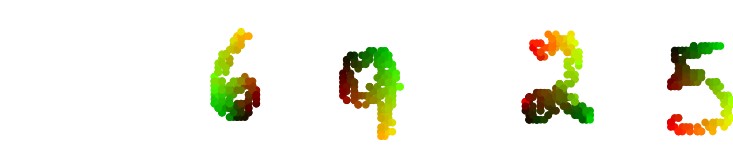

Figure 6: Left image: Matching of the numeral "6" to the numeral "9". Right image: Matching of the numeral "2" to the numeral "5". The colors represent which point in one numeral corresponds to another point in the other numeral.

