# OpenReview forum: "Globally solving the Gromov-Wasserstein problem for point clouds in low dimensional Euclidean spaces"
_NeurIPS.cc/2023/Conference — NeurIPS 2023 poster_

### Official Review · Reviewer_xsEn · 2023-06-26

**Soundness:** 3 good
**Presentation:** 3 good
**Contribution:** 3 good
**Rating:** 6
**Confidence:** 4

**Summary:**

This paper consider the Gromov-Wasserstien (GW) problem with quadratic cost, a (non-convex) quadratic optimization problem over the space of probability measures (in this work, restricted to uniform discrete measures supported on $n$ points) which takes the form (in discrete setting):
$$\min_{\Gamma} \sum_{1 \leq i,j \leq n} ||x_i - x_{i'}|^2 - |y_j - y_{j'}|^2|^2 \Gamma_{ij} \Gamma_{i'j'},$$
where $\Gamma$ is a doubly stochastic matrix of size $n \times n$.

This problem is known to be hard to solve globally. This paper proposes an approach to do so that is tractable when the point clouds $X= (x_1,\dots,x_n)$ and $Y = (y_1,\dots,y_n)$ are in low dimension---say $d$ [edit: fix rendering]. The core idea is that the GW problem can be reparametrized as

\begin{equation}\tag{1}
\min_{(W,w), \Gamma} - |W|^2 - w + c_0,
\end{equation}
where $\Gamma$ is still a bistochastic matrix, $W,w$ must satisfy the relation $W = 2 X \Gamma Y^T$ and $w = \braket{L,\Gamma}$, where $L$ and $c_0$ are constant term (wrt $w,W,\Gamma$). Note that $W$ can be understood as the correlation between $X$ and $Y$ wrt the joint law $\Gamma$, which is of size $d \times d$, so losely speaking, the quadratic part of the GW problem does not depend on $\Gamma$ (which is of size $n \times n$) but only on the correlation it induces (which is a much smaller object when $d$ is small).

From this reformulation, the idea of the paper to globally solve GW is the following:
- Let $\mathcal{F}$ denote the constraint polytope which links $W,w$ and $\Gamma$. This space is too complex to be represented explicitly (by linear constraints).
- Build a bounding box of $\mathcal{F}$. This is doable because we know that the correlation matrix between $X$ and $Y$ has (explicit) bounds on its entries---for instance when $d=1$ this is simply the cost of the increasing (resp. decreasing) matching.
- Build iteratively a sequence of "lower approximation" $(H_N)_N$ of $\mathcal{F}$ (made of supporting hyperplanes of $\mathcal{F}$), in sense that minimizing the functional in (1) over $H_N$ yields a lower bound $L_N$ for the GW problem. Note: this is a concave minimization problem, which is tractable in low dimension ($d \times d + 1$ here).
- Let $(W_N, w_N)$ be the solution on $H_N$. Typically $W_N,w_N \not\in \mathcal{F}$ (otherwise, we have found a global minimizer of GW), but from this we can build a new constraint $H_{N+1}$ through solving a standard OT problem, providing a doubly stochastic matrix $\Gamma_N$ which is by definition sub-optimal, hence giving an upper-bound $U_N$ for GW.
- Eventually, the authors prove that $U_N - L_N \to 0$ as $N \to \infty$, so the proposed approach yields a practical (in low-dimension) algorithm that converges to a **global** solution of the GW problem.

**Strengths:**

The paper considers the difficult (and important, in my opinion) question of globally solving the GW problem.
The proposed approach is based on existing ideas (parametrization of GW by low-dimensional matrices) but pushes them further to get an original and interesting approach. Having globally converging algorithms for GW, even if restricted to low dimensional setups with "not-so-many points", can be useful to assess the quality of other algorithms that may only converge locally (maybe some of them still converge globally "most of the time", etc.).

The paper is clear and does not sacrifice mathematical technicality. Proofs of Prop 2 and Thm 1 have been checked and no major flaw was identified (aside from small details).

I also appreciate that the paper immediately acknowledges that its approach is limited to low-dimensional problems (which is not a major issue; using GW in low dimension is completely natural).

**Weaknesses:**

# 1. On the convergence of the algorithm

From my understanding, the proof of global convergence of the algorithm (Theorem 1) is purely asymptotic: the gap $\epsilon_N = |U_N - L_N|$ is controled by $|\theta_N - \theta_{N+k}|,\ (\forall k)$ where $\theta_N = (W_N, w_N)$ is compactly supported, which implies by contradiction that $0$ must be the single accumulation point of $\epsilon_N$.

While this makes sense (up to few technical considerations, see below), this can be considered as a weakness (at least from a theoretical viewpoint): the convergence of $\epsilon_N \to 0$ is controlled by "how fast $\theta_N$ accumulates", and even in low dimension, without further investigation (i.e. thinking that $\theta_N$ moves arbitrarily in a compact set), it may take a lot of time to reach low gap $\epsilon$ (and this seems to be suggested by the experiments, where the criterion is set to $10^{-8}$ when $d=2$, but to $10^{-2}$ when $d=3$).

I am not saying that this is what happen in practice, nor that this invalidates the contribution of this theoretical result, but a discussion on this may be welcome.


# 2. On numerical experiments

While the proposed experiments are conducted in a reasonable way, they remain somewhat limited in my opinion. In particular,
- [related to the point above] I would have appreciated to have more illustrative experiments on the algorithm behavior, its convergence, etc.
- As far as I can tell, the paper only compares with the work "local search" [16] (Peyré et al., 2016). Why not comparing with more modern works, as [17] (Scetbon et al., 2022), [D] by Sejourné et al. (2021) or [E] (Li et al., 2023)? Note that [17] seems to handle larger instances ($n=10^5$ points), but may fail to globally converge as far as I can tell. Showcasing the strengths of the current work vs [17] (probably the closest work in its spirit), even in illustrative scenarios, would be of interest.



# Minor comments (rather suggestions than actual weaknesses)

1. In Algorithm 1, the upper bound is updated as $U_{N+1} \leftarrow \min(U_N, \text{OT cost} )$. From my understanding, this makes $U_N$ non-increasing, while $L_N$ is increasing (as a minimization problem with more and more constraints), so that $\epsilon_N$ is decreasing. This is never mentioned as far as I can tell; but it seems to be used in the proof when saying that "if $\epsilon_N \not\to 0$, then it must be lower bounded". Am I correct? Also, the proof says "assume that $\epsilon_N = \dots$" (which to me means "assume that $U_{N+1} = \text{OT cost}$ rather than $U_{N+1} = U_N$"), but never discusses the possibility that $U_{N+1} = U_N$. In any case, this does not invalidate the proof as, from my understanding, the $\min$ in the algorithm is not required to prove convergence (even if we cannot ensure that $\epsilon_N$ is decreasing). But in any ways, I think that this is worth some clarification.

2. The discussion related work and context can be improved. For instance, I do not think that [16] is a suited reference in the introduction with mentioning the GW problem for the first time. It would rather be credited to either Mémoli (2011), or Sturm (2012 - "The space of spaces"). What is new vs known (from [16], or for instance from [A, Sec 2.2.3] and related works) in sections 2 and 3 should also be highlighted. Similarly, using that the quadratic term in GW only depends on $X \Gamma Y^T$ is also used (among other) in [B, C] (note : [C] is a preprint put on arxiv after neurIPS submission, this is a suggestion for the revised version, not a criticism).

3. [typo] line 90 : "has" should be "have" I think?

4. [typo] line 114, I think that "maximum" should be "minimum" ?

5. I wonder how useful are the variables $Z_N, \alpha_N$, given that $Z_N$ is simply $2 W_N$ and $\alpha_N = 1$. I understand that this is a convenient way to write a "general" hyperplane equation $\braket{Z,W} + \alpha w \leq \beta$, but to me it turned out to hinder the reading a bit.

6. [typo] line 168, "that the that".


# References

- [A] A contribution to Optimal Transport on incomparable spaces, T. Vayer, 2020
- [B] On the existence of Monge maps for the Gromov-Wasserstien problem, Dumont et al., 2023.
- [C] The Gromov-Wasserstein distance between spheres, Arya et al., 2023.
- [D] The Unbalanced Gromov Wasserstein Distance: Conic Formulation and Relaxation, Sejourné et al., 2021
- [E] A Convergent Single-Loop Algorithm for Relaxation of Gromov-Wasserstein in Graph Data, Li et al., 2023

**Questions:**

I think that the work may be improved by adding:
- A discussion / numerical illustration on the convergence (rate) of the algorithm (Section 1. in Weaknesses), or an explanation of why this is not relevant.
- A more precise comparison with concurrent works, in particular [17] (which is the closest to the current work as far as I can tell), from both a theoretical and numerical perspective ; where scalability but also quality of the result (which, I guess, may favor the proposed approach). If the comparison is not meaningful, please explain why.

**Limitations:**

The authors clearly state that the work is dedicated to low-dimensional point cloud, which is its main limitation.

Aside from that, I do not see any potential societal impact specific to this work.

---

> ### Author Rebuttal · Authors · 2023-08-07
>
> Thank you sincerely for your comments and questions. See below for our answers.
>
> Weaknesses:
> On the convergence of the algorithm:
>
> We are working on quantifying the convergence rate. The number of iterations are bounded by  ($O((1/\epsilon)^{\ell_x\ell_y+1})$), to cover the compact set, but the low dimensional QP problem is increasing in complexity for every iteration. The experiments in the supplementary material 1.4.1 suggest much faster rates, but this is work in progress.
>
> If the problem is extremly symmetric, e.g. equidistant points on the surface of a sphere are being matched to equidistant points on another sphere, then the convergence would be very slow. We have not introduced symmetry breaking concepts into the scope of the method yet.
>
> On numerical experiments:
>
> 1. We are planning to add another experiment matching geometrical data e.g. MNIST objects, where we can highlight issues with symmetry using local methods.  In brief, comparing two figures in MNIST takes in mean 0.7 seconds (standard deviation 0.39 seconds) running in mean 64 iterations (standard deviation 24 iterations) on problems of size 169 points in mean (standard deviation 30 points). In comparison [16] has a mean relative error of about 10% in its measurements of the GW discrepancy which for some figures are more than the mean gap between the different classes.
>
> 2. Indeed, the result in [17] and [D] increase the performance solving the linear OT, which can be incorporated in our algorithm as well. However the main purpose of this paper is to show a simple method to globally solve the GW problem with accuracy certificates. Since we have not optimized the subparts of our algorithm using the computational methods from, e.g., [17], we thing that is makes more sense to compare with [16]. Regarding scalability, note that our method fully decomposes the computations in terms of non-convex low dimensional quadratic problems and linear OT problem, thus one can apply any of the modern tricks on the OT problems in order to improve speed and problem sizes. This has not been a focus of this work and we have just used of the shelf solvers for the OT problem in order to simplify the description and implementation.
>
> Minor comments:
>
> 1. Indeed, the upper bound $U_N$ is the least upper boud, which decreases, this the gap is monotonically decreasing. This can be stated more clearly.
>
> 2. Indeed, this is an error that occured during redisposition of the manuscript. To our understanding, the Gromov-Wasserstein distance should be credited to Mémoli [13] or the 2007 proceedings article. Thank you for pointing out the newly published work!
>
> 3-6. Indeed, thank you for finding the typos.
>
> Questions:
>
> 1. We have answered this under section weaknessess.
>
> 2. The proposed method considers the topic of a global optimum of the GW distance. The main point of the paper is to show that one can find a global optimal solution within a reasonable time, and we do not claim to be faster than [16] or [17] on any GW-problems. The local search algorithms converge faster to a local optimum, but the advantage of our approach is that we are guaranteed to reach a global optimum. For many applications, this can be critical.
> Also note that we have not focused on optimizing the computational complexity in each of the subparts of the algorithm. For example, we do not utilize Sinkhorn’s method for solving the optimal transport subproblems (Equation 9). It is true that [17] utilizes low rank structures similar to the ones we consider for improving the computational complexity of [16]. Similarly, we could use the methods from [Scetbon, Cuturi, Peyré - ICML, 2021] for improving the computational speed of the subproblems in our method. But since we have not optimized these computations, we think that it makes more sense to compare with the method in [16]. We will add this comment to the final version of the paper to motivate why we compare with [16] instead of [17].

---

> > ### Comment · Reviewer_xsEn · 2023-08-12
> > **Thanks**
> >
> > Thank you for taking time to answer my question. I am fairly convinced by the point
> >
> > > Since we have not optimized the subparts of our algorithm using the computational methods from, e.g., [17], we thing that is makes more sense to compare with [16].
> >
> > and as such will keep my supportive grade.

---

> > > ### Author Response · Authors · 2023-08-18
> > >
> > > Thank you!

---

### Official Review · Reviewer_yHuE · 2023-06-29

**Soundness:** 3 good
**Presentation:** 3 good
**Contribution:** 3 good
**Rating:** 6
**Confidence:** 4

**Summary:**

The Gromov-Wasserstein optimal transport problem is a non-convex problem known to be hard, closely related to QAP. The authors consder its special case when the two involved metric spaces are Euclidean with small numbers of dimensions. We want to permute one set of points such that the sum of squares of differences of Euclidean distances between all point pairs is minimized. This simplification of this problem (compared to the general GW OT problem) is possible because the matrix of Euclidean pairwise distances for any set of points in $R^k$ has rank at most $k+2$. This allows the authors to formulate the problem as minimization of a simple concave quadratic function of a small ($kl+1$ where $k,l$ are the dimensions of the spaces) number of parameters over a convex polyhedral feasible set, which is a projection of the set of doubly-stochastic matrices. This problem is solved to global optimality by alternating two steps: (1) globally solve the concave minimization problem over an outer approximation of the feasible set, (2) improve this outer approximation by generating a cutting plane (a linear inequality valid for the above projection of the set of doubly-stochastic matrices). Step 1 can be done either by an off-the-shelf concave minimization algorithm (such as branch&bound) or by a reasoning over the set of extremal points of the feasible set (which is fast for low-dimensional spaces). Step 2 leads to solving the ordinary linear (Kantorovich) OT problem. The algorithm is proved to converge to a global optimum.

The proposed algorithm is first tested on synthetic data. Here, we match two randomly generated (either uniformly on a disc or normally distributed around the origin) point sets from $R^2$ or $R^3$. This is compared to other globally optimal methods solved by Gurobi and to the local search method [17] with multiple initializations. Second, the method is tested on a real application from biology. These experiments show that the method is almost always much faster than the other methods (the global methods are in fact usable only for small instances, due to the large number of parameters).

**Strengths:**

A simple algorithm for a difficult problem, without many tuning parameters, which performs well in experiments.
Clearly written.

**Weaknesses:**

The most important issue is limited experimental testing. The main experiment is done on synthetic, randomly generated data. However, it is known that optimization algorithms often behave very differently on random data and on real data.

Moreover, I do not quite understand the setup in the biological experiment, especially I get lost in the 1st paragraph in section 5.1. Please explain better. What is $n, l_x, l_y$ in this case?

To show more clearly the strengths and weaknesses of the approach, it should be tested on as many instance types as possible. Currently I am not convinced that the method would not last very (unacceptably) long for some instances from practice.
One option is to make more extensive synthetic experiments, with more complex data than just uniformly/normally distributed. E.g., one can take a non-random set of points in 2D or 3D (draw a shape or use a shape from a public shape database) and to synthetically generate the second set by permuting these points and adding noise to them (gaussian, uniform) or replace some of them with outliers.

Another obvious suggestion is shape/object matching on real data, where the point distances are measured by Euclidean metric (rather than geodesic). Possible inspirations for such experiments is [13, 17] (but, I believe, also other works).

As for novelty: Although the method just combines tools well-known in optimization, the main idea (decoupling the concave minimization part and the cutting plane part) is clever. Unfortunately, I do not know the relevant literature well enough to be sure that a similar approach has not been proposed before. I am surprised that no globally optimal algorithm for globally solving this problem has been proposed before by others - such algorithms have been apparently proposed only for general GW problem and QAP.

The initial formulation (1) of the GW problem seems to be wrong because, to my knowledge, the GW optimal-transport problem optimizes over doubly-stochastic matrices rather than permutation matrices (i.e., it allows "soft matching"). Therefore, the relaxation from (4) to (5) is unnecessary and in fact misleading. If (5) minimizes a concave function over a convex set, the two problems are indeed equivalent, but the explanation should go the other way.

Minor issues / suggestions:

- 77: "the first two sums" should be "the 1st and 3rd sum"

- In (2), it would be logical to omit the 2nd and 3rd terms because they do not depend on Gamma (as noted above).

- 92: You may wish to justify/cite why a Euclid distance matrix has rank at most $l_x+2$.

- Proposition 1 should not be really a proposition, it is just a algebraic manipulation.

- 133: The part defining the set $\mathcal{F}$ is over-complicated and confusing. Why don't you write just
$\mathcal{F} = \\{ (2X\Gamma Y, \langle L,\Gamma\rangle) \mid \Gamma \in P \\}$, which shows that $\mathcal{F}$ is a linear map of $P$.

- In the final version, Table 1 should report also the number $N$ of iterations (= added cutting planes) needed to achieve the prescribed accuracy. Currently, this is only in the supplement.

**Questions:**

Why were experiments on more real instance types not included, such as with shape/object matching? Is there a substantial obstacle or you just considered this as out of the scope of this paper?

Can the method be extended to be resilient to outliers? One option for this would be to use 1-norm rather than 2-norm in pairwise distances - but this would violate the low-rank assumption and make the approach inapplicable.

Can the trick be applied to a wider class of problems, such as a wider subclass of QAP?

**Limitations:**

The experiments are limited, not convincingly showing efficiency of the method on a wide enough class of real instances.

---

> ### Author Rebuttal · Authors · 2023-08-07
>
> Thank you sincerely for your comments and questions. See below, for our answers.
>
> Weaknesses:
>
> Limited experimental testing:
>
> The main focus of this paper is on difficult problems that appear in, e.g., computational biology and where there are symmetries in the data and thus there is a large set of local optima. For many problems with image matching on simple figures (cats, dogs, etc.), both local search methods methods and the proposed global optimization method would be very quick. The local search would typically give the correct result and be quicker, but without any guarantees. Our method would be slower, but still fast, and come with guarantee. However, it makes sense to add some additional such examples, and illustrate that our method works also for this case. Thus we will include additional numerical simulations on known simple geometrical objects, such as MNIST. In brief, comparing two figures in MNIST takes in mean 0.7 seconds (standard deviation 0.39 seconds) running in mean 64 iterations (standard deviation 24 iterations) on problems of size 169 points in mean (standard deviation 30 points). In comparison [16] has a mean relative error of about 10% in its measurements of the GW discrepancy, which for some figures are more than the mean gap between the different classes
>
> Setup in the biological experiment:
>
> In the biological experiment $n$ is the number of points, $\ell_x$ is the dimensionality of the space which the points in $X$ is located in and $\ell_y$ accordingly. In the biological experiment in section 5.3: $n=500$ and $\ell_x=\ell_y = 2$ (since we compare 2-d images). This shall be included in the text. Thank you for pointing this out.
>
>
>
> Euclidean metric: By using the Euclidean metric rending a high rank on the distance matrices with mixed positive and negative eigenvalues, such problems falls outside the class for which our paper considers. We are currently working on extending the results in this direction.
>
> GW problem: The original Gromov-Wasserstein description [13] indeed relax the problem to general marginals, and we shall add this in the introduction before Equation (1) to correct the terminology. Thank you for pointing this out. When the marginals are uniform over n points the result is still valid with permutations as the resulting matching, and closely related to the pre-model using couplings and correspondences in the Gromov-Hausdorff distance. One may consider discrete marginals $\mu$ and $\nu$ in this model. Then, the only difference is that $L = (1^T\mu + 1^T\nu) m_xm_y^T -4m_x\nu ^T Y^T Y-4X^T X\mu m_y^T$ and $c_0 =(\langle C_x,C_x\rangle+ \langle C_y,C_y\rangle -4\nu^Tm_y \mu^Tm_x)/2$.
>
> The minor issues:
>
> line 77: Indeed, thank you for pointing this out!
> Eqn (2): We decided to keep this as it is relevant for the discrepancy, even though not relevant for the optimization problem.
> Line 92: We can add this close to (3) where this is clear by a sum of matrices with rank 1, $\ell_x$ and 1.
> Proposition 1. Yes, but we added to increase readability and clarity.
> Line 133. Thank you for this suggestion!
> Table 1. Yes, we shall add the number of added cutting planes to the tables.
>
> Questions:
>
> We realize that there is a gap between the random data and the biologic example. We are planning to add a low dimensional geometrical data examples e.g. MNIST to show the performance and when symmetry becomes the key issue for local search methods. Such data is in scope for the method indeed.
>
> In order to handle ouliers and 1-norm distance matrices we are violating the low rank assumption indeed. Currently such problems are not within the problem class considered in the paper, but we are looking to generalize the framework for handling such problems.
>
> As the QAP problem is restricted to permutations, it would be a natural extension to consider a wider class of matrices. When the matrices contain both positive and negative eigenvalues, this becomes a much harder problem, which is currently out of scope of the method, but we are looking into handling such problems.

---

> > ### Comment · Reviewer_yHuE · 2023-08-15
> >
> > Thank you very much for your rebuttal. I find (along with the other reviewers) the algorithm clever and nice, which will probably ensure acceptance. However, I still believe it is a pity that you did not include a wider range of data types in the experiments. Your explanation why you did not do this is that "The main focus of this paper is on difficult problems that appear in, e.g., computational biology and where there are symmetries in the data and thus there is a large set of local optima." I find this explanation unfair because there are several other scenarios with symmetries in the data (and hence many similar local minima), such as: shape (represented by point cloud) matching where the shape has symmetries, or feature matching in stereo images with repeated patterns (e.g., matching two views of a building with many similar windows). These experiments could be on both synthetic and real data. Including such experiments might ensure a greater impact of the paper. (I find your planned MNIST experiment insufficient in this respect.)
> >
> > However, I find the paper acceptable even without these additional experiments. Based on other reviews and rebuttals, I increase my rate to "weak accept".
> >
> > One more suggestion (you may ignore it): some reviewers complain about missing convergence rate analysis. This is somewhat (weakly) related to the hardness of the problem solved. Clearly, the  QAP problems and its low-rank version are NP-hard. But how about approximability? Perhaps, low-rank QAP is easier to approximate than the general QAP.  Is this known? If the low-rank QAP has no FPTAS (which I assume), the convergence rate cannot be polynomial in problem size and $1/\epsilon$. Perhaps, a remark on this might be useful.

---

> > > ### Author Response · Authors · 2023-08-18
> > >
> > > Thank you for the reconsideration of the score!
> > > We will include a more detailed discussion about the convergence rate.
> > > The shape examples used in [16] (https://github.com/gpeyre/2016-ICML-gromov-wasserstein/tree/master/code/data/shapes), has a mean relative error of 3% using [16] compared to the global result, when sampling 500 points from the images.

---

### Official Review · Reviewer_FaYe · 2023-07-03

**Soundness:** 4 excellent
**Presentation:** 4 excellent
**Contribution:** 4 excellent
**Rating:** 8
**Confidence:** 4

**Summary:**

- The paper introduces a novel algorithm for efficiently computing the Gromov-Wasserstein (GW) distance for low-dimensional point clouds. The proposed method offers a transformative approach by transforming the GW distance problem into a sequence of concave optimization problems over convex sets, thereby improving computational efficiency.

- By leveraging this new algorithm, the authors conducted comparative evaluations against existing techniques for computing the GW distance. The results demonstrate significant improvements in terms of computational speed.

**Strengths:**

- One of the key strengths of this paper is the introduction of a novel and interesting algorithm. The proposed method offers a fresh perspective on computing the Gromov-Wasserstein (GW) distance for low-dimensional point clouds.
- The research holds significant importance in the field of machine learning and data science, given the growing popularity of the GW distance. By proposing a method that accelerates the computation of the GW distance, the paper addresses a practical need and offers a valuable contribution to the field.
- The clarity of presentation is a notable strength of this paper. The authors effectively communicate their ideas, methodologies, and findings, making the paper accessible to readers.
- The experiments well support the acceleration effect of the proposed algorithms.

**Weaknesses:**

A potential weakness of the paper is the lack of a complexity analysis for Algorithm 1, which could be addressed to further enhance the paper.

**Questions:**

- line 190: in the last inequality, am I right that a square root is missing?
- line 196: Is there a typo in the lower bound?

---

> ### Author Rebuttal · Authors · 2023-08-07
>
> Thank you sincerely for your comments, feedback, and questions. See below for our answers.
>
> Weaknesses:
>
> We agree that this is a weakness and we are currently working on quantifying the convergence rate. For a given dimensionality and number of points, it should be straightforward to show that our algorithm finds an $\epsilon$ accurate solution in $O((1/\epsilon)^{\ell_x\times \ell_y+1})$ iterations. However, this trivial complexity bound is not very satisfying and it doesn't take in consideration the increasing complexity of the QP problem. We believe that it should be possible to prove a faster convergence rate. In practice, we have experienced that the method converges much faster as exemplified in section 1.4.1 in the supplementary material.
>
> Questions:
> 1. Line 190: Yes, thank you for finding this error.
> 2. Line 196: The notation might be confusing, what we mean is that the lower bound is limited to the largest possible $\|W\|_F$ obtained in the bounding box. We will improve the readability of this.

---

> > ### Comment · Reviewer_FaYe · 2023-08-19
> >
> > Thank you for answering my questions. After reading the rebuttal and other reviews, I would like to maintain my current score.

---

### Official Review · Reviewer_G7H7 · 2023-07-06

**Soundness:** 3 good
**Presentation:** 3 good
**Contribution:** 3 good
**Rating:** 6
**Confidence:** 3

**Summary:**

This paper solves the Gromov Wasserstein (GW) distance problem for squared Euclidean norm by considering a low-dimensional space on which the computation is performed, using a cutting-plane method. To this end, they write the GW problem as a low-rank optimization problem. Their algorithm is supported by theoretical convergence guarantees.


**Strengths:**

This article makes it possible to compute the Gromov-Wasserstein distance efficiently. In a way, this paper extends the results of [17] which reformulated the Gromov-Wasserstein problem as a low-rank quadratic problem. The idea of using the cutting plane algorithm to efficiently solve this optimization problem in a projected low-dimensional subspace is new, and the proposed algorithm converges to a global optimal solution.

**Weaknesses:**

Solving problem (9) at each iteration of the algorithms remains computationally heavy, as it corresponds to solving an optimal transport problem with $\Gamma$ a $n\times n$ matrix.

For a more relevant contribution, and as the use of the low-rank quadratic structure of the Gromov-Wasserstein optimization problem has already been exploited in [17], the authors should compare their results with [17] in experiments, and not only with the local search method of [16].

**Questions:**

The matrices $C_x$ and $C_y$ are assumed to be positive definite and of low rank. How can the low-rank assumption be ensured? What does this mean in terms of the point clouds $X$ and $Y$?

Can the cutting plane algorithm trick be extended to the case where both point clouds have weights (i.e. a non-uniform discrete measure), and for which the probability measures do not have the same number of points (therefore the mass in the transport plan $\Gamma$ could split)?

**Limitations:**

A comparison with more recent methods than [16] for computing the Gromov Wasserstein distance should be included.

---

> ### Author Rebuttal · Authors · 2023-08-07
>
> Thank you sincerely for your comments and questions. See below for our answers.
>
> First we would like to clarify that the paper is not just an extension of the work [16] and [17]. Even though we also consider the GW-problem, our method is guaranteed to reach the global optimum for the problems we consider. This is to the best of our knowledge the first results that guarantees a global optimal solution for a class of GW-problems where the domain is multidimensional (for 1-d such results exists). Also, note that even though the work [17] uses similar low rank structures, the structures are used completely different. In [17], the low rank structures are used to speed up the method from [16]. In our paper, the low rank structure is used to ensure convergence to a global optimum (and not just a local).
>
> Weaknesses:
>
> The proposed method considers the topic of a global optimum of the GW distance. The main point of the paper is to show that one can find a global optimal solution within a reasonable time, and we do not claim to be faster than [16] or [17] on any GW-problems. The local search algorithms converge faster to a local optimum, but the advantage of our approach is that we are guaranteed to reach a global optimum. For many applications, this is critical.
>
> Also note that we have not focused on optimizing the computational complexity in each of the subparts of the algorithm. For example, we do not utilize Sinkhorn’s method for solving the optimal transport subproblems (equation 9). It is true that [17] utilizes low rank structures similar to the ones we consider for improving the computational complexity of [16]. Similarly, we could use the methods from [Scetbon, Cuturi, Peyré - ICML, 2021] for improving the computational speed of the subproblems in our method. But since we have not optimized these computations, we think that it makes more sense to compare with the method in [16]. We will add this comment to the final version of the paper to motivate why we compare with [16] instead of [17]
>
> Questions:
> 1. With Squared Euclidean distance, this is ensured trivially as $C_x$ and $C_y$ will have rank at most $\ell_x+2$ and $\ell_y+2$. For more general settings, it gets a bit more complicated but this is out of scope of this paper. Work on low rank approximations of point clouds to their distance is a whole separate field of investigation, which we can add references to.
> 2. Yes, then $L = (1^T\mu + 1^T\nu) m_xm_y^T -4m_x\nu ^T Y^T Y-4X^T X\mu m_y^T$, $c_0 =(\langle C_x,C_x\rangle+ \langle C_y,C_y\rangle -4\nu^Tm_y \mu^Tm_x)/2$ where $\mu$ and $\nu$ are the marginals.
>
> Limitations:
>
> Please see the answer in the weaknesses section.

---

> > ### Comment · Reviewer_yHuE · 2023-08-15
> >
> > Let me just remark that the sentence "This is to the best of our knowledge the first results that guarantees a global optimal solution for a class of GW-problems where the domain is multidimensional" may be somewhat naive. Indeed, there are such algorithms - e.g., exhaustive search over all permutations or (spatial) branch-and-bound. What you probably meant is "in reasonable time in practice".
> > (Note, I am a different reviewer.)

---

> > > ### Author Response · Authors · 2023-08-18
> > > **Answer to Reviewer yHuE**
> > >
> > > Indeed, you are completely correct about this. Thank you.

---

> > ### Comment · Reviewer_G7H7 · 2023-08-16
> > **Response to the authors**
> >
> > Thank you for your detailed reply and for your intention to add to the experiments section one with the real-world MNIST dataset. From the answers you have provided to the various reviewers, the efficient global optimization approach you propose for GW computation seems more innovative than at first sight, and in particular the reason for comparing your method with [16] is more convincing. As the presentation of the general idea and theoretical parts are, in my opinion, particularly clear, I will increase my score from 4 to 6.

---

> > > ### Author Response · Authors · 2023-08-18
> > > **Answer to Reviewer G7H7**
> > >
> > > Thank you for your reconsideration!

---

### Official Review · Reviewer_pM3y · 2023-07-06

**Soundness:** 3 good
**Presentation:** 3 good
**Contribution:** 4 excellent
**Rating:** 6
**Confidence:** 4

**Summary:**

This paper proposes a new algorithm to solve the Gromov-Wasserstein problem between two sets of points in Euclidean spaces when the ground cost is the squared Euclidean norm. This is done by first reformulating the Gromov-Wasserstein problem as a low-rank QAP problem,
then relaxing the set of admissible couplings in a way that the optimal solutions of the obtained relaxed problem stays the same, and then using a cutting-plane method to solve the relaxed problem. The obtained algorithm provides at each iteration a lower bound and an upper bound on the value of the optimal solution, and it can be proven that the gap between those bounds theoretically converges to zero at infinity, implying that the proposed algorithm theoretically converges to a global optimum. The experiments offer a comparison in term of computational efficiency between the proposed algorithm and a more "traditional" algorithm, and emphasize the importance of converging to a global optimum when using Gromov for applied problems.

**Strengths:**

I enjoyed reading this paper. To the best of my knowledge, the proposed algorithm is new.
Beside the proposed algorithm in itself, the paper highlights that most of algorithms which compute Gromov only converge to a local minimum, which is a problem that is often ignored in the literature. Moreover, when solving the
Gromov-Wasserstein problem with "traditional" algorithms, it is not even possible to know in practice whether the solution
we've converged to is a local or global optimum. Thus, proposing an algorithm that is guaranteed to converge towards the global optimum is a real strength in my opinion.

**Weaknesses:**

- As someone familiar with the Gromov-Wasserstein problem but not expert in optimization, I find this paper a bit difficult
to read because some important steps in the mathematical reasoning are lacking of details and so, in my opinion, too much of the reasoning is left to reader.
- I think the experiment section could be improved: the experiment of Figure 2 could be clearer, and it could be nice
to have a synthetic experiment that exhibit a case where the proposed algorithm converges to the global optimum whereas the local search method remains stuck in a local optimum.


**Questions:**

The following questions are related to the steps of reasoning that I find could be clearer:
- 1/ What is the definition of the rank of a QAP problem?
- 2/ Why do we need to relax Problem (4) to doubly stochastic matrices instead of permutation matrices before applying
the cutting-plane method?
- 3/ Can you explain how to obtain the initial $ (Z_r,\alpha_r,\beta_r)_r $ from solving the bounding box problem?
- 4/ I'm a bit confused about Equation (8a): are we looking for a different $ \Gamma $ for each $ i,j$? The way it is written suggests we're looking for the same for all $ i $ and $ j $.
- 5/ $ N $ stands for the current iteration number,  but also for the number of initial constraints. I find that confusing.

Experiments:
- 1/ In Figure 2 left: are the differents structure plotted are local optima obtained using a local search method? I'm not sure
to understand Figure 2 right either: the blue is the result of your algorithm? can you explain why it is better than the result with the local search method and where do we see the comparison with the expert evaluation?
- 2/ How the methods compare itself (in terms of computational efficiency) to approached methods (as for instance Entropic Gromov, sliced Gromov,etc) with several random initializations?


**Limitations:**

The authors has not clearly discussed the limations of the algorithm.
In my opinion, the principal limitation is that the method remains computationally costly, and so, as this is the case for almost all methods that compute Gromov, it is still unusable in higher dimension or when the number of points is large.

---

> ### Author Rebuttal · Authors · 2023-08-07
>
> Thank you sincerely for the comments, feedback, and questions. See below for our answers.
>
> Weaknesess:
> 1. We will go over the mathematical derivations again and try to improve the readability for the final version of the paper.
> 2. We are planning the expand the numerical results in the final version. We will improve the description of the experimental setup in Figure 2. In the supplementary material 1.4.3, we presented the statistical performance of the local search method on this example, which in mean has a relative error of above 10%. We will move this into the paper in the final version. We also plan to change Figure 1 so that it illustrates that one could get stuck in a second local optimum using a local method. Here we can identify the whole region of attraction for the local optimum that is not a global optimum.
>
> Questions:
> 1. A QAP problem may be written on the standard form $\min_{\Gamma \in P} trace(W\Gamma D\Gamma^T)$. Now $ trace(W\Gamma D\Gamma^T) = vec(W\Gamma D)^T vec(\Gamma) = vec(\Gamma) ^T (D^T\otimes W)^T vec(\Gamma)$. Then we denote the rank of $D^T\otimes W$ as the rank of the QAP. In the GW case using the squared Euclidean norm we have the quadratic expression $trace (X^TX \Gamma Y^TY \Gamma^T)$. If we identify $W = X^TX$ and $D = Y^TY$, then the rank is the product of the dimensionality $l_x$ and $l_y$ of $X$ and $Y$ respectively, because of the eigenvalue structure of the kronecker product.
> 2. We are not aware of any convenient way to represent the set of points that corresponds to permutations in the restricted low dimensional space. The main purpose of the relaxation is to obtain easier subproblems, and to the best of our knowledge, optimizing over permutation matrices is in general a difficult combinatorial problem.
> 3. Equation 8 describes the bounding box problem, as they describe the min and max values of the elements in the low rank representation. Equation 8a are $\ell_x\ell_y$ equations indexed by $i = 1... \ell_x$ and $j = 1...\ell_y$ giving $Z_{i,j} = \delta_{i,j}$, $\alpha_{i,j} = 0$ and $\beta_{i,j} = max_\Gamma (2X\Gamma Y^T)_{i,j}$ for the maximum expression for example.
> Note also that these inequalities can be written more plainly as $W^{\min}\le W\le W^{\max}$
> where $W^{\min}$ and $W^{\max}$ are specified elementwise by the minima and maxima in Equation 8a.
> 4. Equation 8a  is $\ell_x\ell_y$ equations with one optimal $\Gamma$ for each $(i,j)$.
> 5. N stands for the number of constraints. We will clarify this and how it relates the iteration number in the final version.
>
> Experiments:
> 1. The proposed algorithm provides a sequence of permutations that may be close to locally optimal as visualized in figure 2 left. The objective value in these permutations may be very close to the global optimum (written in relative error) but suggest a permutation which is very far away from the global optimum permutation geometrically, here being rotations and mirror symmetries. This is only to exemplify the complexity of the problem to be solved. Figure 2 right shows classification quality metrics compared to the expert classification (ground truth), i.e., 1 would be a perfect result. The figure shows that the proposed method results in a better solution with regard to all the quality metrics. We will try to clarify this better in the final version.
> 2. A key advantage with this method is that one knows when optimum is reached. When considering local search (e.g., Entropic Gromov) it is difficult to know how close the best solution is to the optimal solution. In particular, this is the case when considering problems with symmetries, which are common in for example computational biology. In the final version, we will more clearly highlight the advantages of our global optimization approach compared to local search with multiple initializations. The sliced Gromov-Wasserstein version handles problems when the spaces are of different nature. However having a distance matrix, under certain conditions it is possible to find a representation in Euclid coordinates.
>
>
> Limitations:
>
> It is true that the method is computationally costly if the number of dimensions increases. However, for small dimensional problems (e.g., 2d and 3d objects which are common in practice), the method scales well in the number of points (in contrast to most other methods for solving GW problems).

---

> > ### Comment · Reviewer_pM3y · 2023-08-15
> > **More precisions and suggestions**
> >
> > Thank you for your clarifications that are quite helpful. I'm fairly conviced that this paper makes a significant contribution in proposing
> > an algorithm for solving the GW problem which converge to a global optimum or which indicates how far we are from it. Yet, I still think this paper could be improved on the presentation of the theoretical results and in the choice of experiments.
> >
> > On the theoretical results:
> > - In my opinion, a justification similar to your answer to question 1/ on the low-rank structure of the problem should definitively be included in the final version since I'm not sure that all readers interested in GW are familiar with the standard form of a QAP.
> > - Same goes for 2/: in my opinion the relaxation should also be justified, even if it is a classic "trick" of optimal transport theory. Furthermore,
> > as Reviewer yHuE  pointed it, this relaxation is misleading because in the classical form of the GW problem, the optimization is performed
> > on the set of double stochastic matrices and not permutation matrices. This is not a big issue in itself, but what makes it misleading is
> > that the whole point of section 3 is to derive Expression (5) that you call "low rank formulation" of the GW problem. Yet, (5) is already known
> > in the literature, see proposition 1 of [16] or section 2.2.3 in [A]. Hence the goal of section 3 is actually to recover a known formula and to point out that this formula has a low rank structure when the cost matrix $ C^x $ and $ C^y $ are squared distance matrix, while the way it is written suggests (in my opinion) that you are deriving new formulas specifically for the algorithm you propose.
> >
> > On the experiments:
> > - A concern I have is that you compare yourself principally with [16] that is basically an entropic-regularized solver of the GW problem, while
> > your proposed algorithm is solving, if I'm understanding correctly, the non-regularized problem. This is especially confusing since
> > [16] is also cited in the doc of Python Optimal Transport for the non-regularized solver of the GW problem. First, I think you should precise
> > more clearly that you're using the Entropic solver and what parameter of regularizations do you use in your experiments. Second, I think
> > you shoot a bit yourself in the foot by not comparing yourself with the non-regularized solver because (i) you only compare yourself with
> > a solver that tackles an "easier" problem, (ii) to the best of my understanding, we cannot conclude currently if in Figure 2 right, the gain of performance is due to converging to a global optimum or to solving the non-regularized problem instead of the regularized one.
> > - I think Figure 2 left and right should be separated in two distincts figures, because there consists in two distinct experiments (although being over the same dataset) and so currently, its difficult to understand what are the goal of these experiments when reading the paper.
> >
> >
> >
> > [A] A contribution to Optimal Transport on incomparable spaces. Vayer, 2020.

---

> > > ### Author Response · Authors · 2023-08-18
> > >
> > > Thank you for you further comments.
> > >
> > > Indeed, proposition 1 should be cited to the literature to the reference you so helpfully provided. We were not aware of this work. Also in the definition of the GW distance we will correct the terminology so that it is clear that the GW problem typically refer to the optimization problem with stochastic matrices (see also answer to reviewer yHuE).
> > >
> > > For comparing with [16], we have used the package provided by their group on Github. Here it is possible to set the regularization parameter to 0, in which case network simplex is used. This is what we are comparing with. We will clarify this in the final version of the paper.

---

> > > > ### Comment · Reviewer_pM3y · 2023-08-18
> > > > **thanks**
> > > >
> > > > thank you for the further clarification. I was not aware that [16] also implemented a non-regularized solver, which explains
> > > > why it is cited by the doc of POT. I've increased my grade.

---

### Author Rebuttal · Authors · 2023-08-07

Dear reviewers,

Thank you sincerely for the work you have put into reviewing the manuscript paper. Your reviews have been thorough, pointing at strengths and weaknesses in the paper and suggested clarifications that benefit the presentation and clarity of the paper. We will do our very best to incorporate the suggestions and remarks under the page limit constraint. In the rebuttal section in each review, we have answered questions and comments in order to make it more clear what answer goes to what question.

Again, thank you for your reviews and feedback.

---

### Decision · Program_Chairs · 2023-09-21

**Decision:**

Accept (poster)

**Comment:**

All the reviews are positive. The paper introduces a novel algorithm for an important problem (GW in Euclidean spaces), leveraging the specific structure of the problem. The rebuttal have helped improved the presentation of the method towards a wider audience. The final version should include a better account for previous works on the same problem, for instance the PhD work of Vayer which already leveraged the concavity of GW between Euclidean spaces. I recommend acceptance.